# Mitochondrial genomic alterations in cholangiocarcinoma cell lines

Athitaya Faipan[1,2], Sirinya Sitthirak[2,3], Arporn Wangwiwatsin[2,3], Nisana Namwat[2,3], Poramate Klanrit[2,3], Attapol Titapun[2,4], Apiwat Jareanrat[2,4], Vasin Thanasukarn[2,4], Natcha Khuntikeo[2,4], Luke Boulter[5], Hasaya Dokduang[2,6], Watcharin Loilome[2,3*]

**1** Department of Biochemistry, Faculty of Medicine, Khon Kaen University, Khon Kaen, Thailand, **2** Cholangiocarcinoma Research Institute, Khon Kaen University, Khon Kaen, Thailand, **3** Department of Systems Biosciences and Computational Medicine, Faculty of Medicine, Khon Kaen University, Khon Kaen, Thailand, **4** Department of Surgery, Faculty of Medicine, Khon Kaen University, Khon Kaen, Thailand, **5** MRC Human Genetics Unit, Institute of Genetics and Cancer, The University of Edinburgh, Western General Hospital, Edinburgh, Scotland, United Kingdom, **6** Faculty of Medicine, Mahasarakham University, Mahasarakham, Thailand

* watclo@kku.ac.th

## Abstract

Cholangiocarcinoma (CCA) is a diverse collection of malignant tumors that originate in the bile ducts. Mitochondria, the energy converters in eukaryotic cells, contain circular mitochondrial DNA (mtDNA) which has a greater mutation rate than nuclear DNA. Heteroplasmic variations in mtDNA may suggest an increased risk of cancer-related mortality, serving as a potential prognostic marker. In this study, we investigated the mtDNA variations of five CCA cell lines, including KKU-023, KKU-055, KKU-100, KKU213A, and KKU-452 and compared them to the non-tumor cholangiocyte MMNK-1 cell line. We used Oxford Nanopore Technologies (ONT), a long-read sequencing technology capable of synthesizing the whole mitochondrial genome, which facilitates enhanced identification of complicated rearrangements in mitogenomics. The analysis revealed a high frequency of SNVs and INDELs, particularly in the D-loop, *MT-RNR2*, *MT-CO1*, *MT-ND4*, and *MT-ND5* genes. Significant mutations were detected in all CCA cell lines, with particularly notable non-synonymous SNVs such as m.8462T>C in KKU-023, m.9493G>A in KKU-055, m.9172C>A in KKU-100, m.15024G>C in KKU-213A, m.12994G>A in KKU-452, and m.13406G>A in MMNK-1, which demonstrated high pathogenicity scores. The presence of these mutations suggests the potential for mitochondrial dysfunction and CCA progression. Analysis of mtDNA structural variants (SV) revealed significant variability among the cell lines. We identified 208 SVs in KKU-023, 185 SVs in KKU-055, 231 SVs in KKU-100, 69 SVs in KKU-213A, 172 SVs in KKU-452, and 217 SVs in MMNK-1. These SVs included deletions, duplications, and inversions, with the highest variability observed in KKU-100 and the lowest in KKU-213A. Our results underscore the diverse mtDNA mutation landscape in CCA cell lines, highlighting

**Data availability statement:** The sequencing data can be accessed at NCBI under the project identifier PRJEB78381.

**Funding:** This work was supported by the NSRF under the Basic Research Fund of Khon Kaen University and Srinagarind Diamond Research Fund (DR63201) to WL. The financial assistance from the Invitation Research Grant (IN67051) and a Postgraduate Study Support Grant of Faculty of Medicine, Khon Kaen University to AF.

**Competing interests:** The authors have declared that no competing interests exist.

the potential impact of these mutations on mitochondrial function and CCA cell line progression. Future research is required to investigate the functional impacts of these variants, their interactions with nuclear DNA in CCA, and their potential as targets for therapeutic intervention.

## Introduction

Cholangiocarcinoma (CCA) is a malignant tumor originating in the biliary tree and can be categorized into three sub-types based on anatomical location: intrahepatic (iCCA), perihilar (pCCA), and distal (dCCA). Perihilar CCA is the most prevalent form of this cancer, comprising 50–60% of CCA cases, followed by distal CCA (20–30%) and intrahepatic CCA (10–20%) [1]. These tumors are the second most common primary liver malignancy, accounting for roughly 15% of all primary liver tumors and 3% of gastrointestinal malignancies worldwide [2]. Notably, northeastern Thailand has the highest global incidence rate of CCA, with 85 cases per 100,000 persons annually [3].

CCA is a complicated and highly heterogeneous cancer distinguished by tumor site, etiology, genetic characteristics, and varied prognostic outcomes. These variables confound much of the analysis of CCA genetics. Cancer cell lines are critical for studying many aspects of tumor biology and therapeutic strategies [4]. Therefore, we leveraged a bioresource of CCA cell lines, including KKU-213A, KKU-100, KKU-055, KKU-452, and KKU-023, which demonstrate differential differentiation abilities to investigate the mtDNA profile of in CCA and provide insight for further prognosis biomarker development in CCA patients.

Mitochondria serve as the energy converters in eukaryotic cells, synthesizing adenosine triphosphate (ATP) through the mitochondrial electron transport chain (ETC) within the oxidative phosphorylation (OXPHOS) system. Mitochondria possess circular double-stranded DNA genomes, termed mitochondrial DNA (mtDNA), comprising 16,569 base pairs. The mtDNA includes a G-enriched inner light strand (L-strand) and a C-enriched heavy strand (H-strand). MtDNA encodes 2 rRNAs and 22 tRNAs for protein synthesis, as well as 13 peptides for ETC and OXPHOS [5,6]. MtDNA has a greater mutation rate than nuclear DNA, estimated to be 10–17 times higher than the nuclear genome due to an absence of complex DNA repair mechanisms [6]. Mitochondria also demonstrate high levels of heteroplasmy, defined as the coexistence of several mtDNA variants within a single cell or organism and which has significant implications for mitochondrial function and disease etiology [7]. A recent study revealed that heteroplasmic variations in mtDNA, particularly single nucleotide variants (SNVs), are associated with an elevated risk of cancer-related mortality in leukemia, highlighting heteroplasmic variation as a potential prognostic indicator for cancer [8].

Oxford Nanopore Technology (ONT) provides long-read sequencing capable of generating sequences thousands of base pairs long. This enables the sequencing of the entire mitochondrial genome (~16.6kb) in a single read. Furthermore, ONT

allows for a more detailed investigation of heteroplasmic deletions inside a single read as well as the identification of complicated, large rearrangements, e.g., duplications, which are difficult to identify using short-read sequencing [9]. Lastly, the study demonstrated the ability of long-read ONT sequencing to detect large-scale deletions and rearrangements in mtDNA, which is important in understanding and diagnosing primary mitochondrial disorders. Hence, nanopore sequencing represents a promising approach for sequencing the whole mitochondrial genome [10] and determining the presence of tumour-associated SNVs.

In this investigation, we conducted comprehensive sequencing of the entire mtDNA using ONT across five distinct CCA cell lines, alongside one non-tumor cholangiocyte cell line, MMNK-1. We then investigated the mtDNA profiles of each cell line to define whether mtDNA contained candidate pathogenic variants within CCA.

## Materials and methods

### Cell lines and cell cultivation

Five *Opisthorchis viverrini* (Ov)-associated CCA cell lines including KKU-213A, KKU-100, KKU-055, KKU-452, and KKU-023, as well as non-tumor cholangiocyte MMNK-1, as listed in the Table 8, were acquired from the Cholangiocarcinoma Research Institute in Khon Kaen, previously documented in the literature [4,11,12]. The cell line collection and study were approved by the Ethic Committee for Human Research, Khon Kaen University (HE671263). Among these, KKU-213A, KKU-100, KKU-055, and KKU-023 were derived from patients with intrahepatic CCA, whereas KKU-452 originated from patients with perihilar CCA. These cell lines were cultivated as a monolayer culture in Dulbecco's Modified Eagle Medium (DMEM) (Life Technologies, Inc.), supplemented with 10% heat-inactivated fetal bovine serum (FBS), 100 U/ml penicillin and 100 mg/ml streptomycin (Life Technologies, Inc.), maintained at 37°C with 5% $CO_2$. In this study, each of the six cell lines had three replicates, a total of 18 samples in each following experiment.

### DNA extraction and mtDNA enrichment

For each of the six cell lines, at least $10^6$ cells were collected for DNA extraction using the QIAamp® DNA Mini Kit (QIA-GEN, Germany) according to the manufacturer's instructions. The DNA samples were pre-treated with Exonuclease V (New England BioLabs, Inc.) to digest any linear nDNA, enrich circular mtDNA, and reduce the risk of nuclear mitochondrial segments (NUMTs) contamination [9]. Each 3 µg of each gDNA sample was treated with 30 units of Exonuclease V at 37°C for 3 h. The reaction was halted by adding 11 mM EDTA and heat-inactivation at 70°C for 30 min. We used 10 ng of circular mtDNA templates to amplify the entire mitochondrial genome using Multiple Displacement Amplification (MDA) and isothermal amplification at 33°C with REPLI-g Midi DNA Polymerase for 8 h., following the protocol outlined by the manufacturer of the REPLI-g® Mitochondrial DNA Kit (QIAGEN, Germany). The amplified DNA was then quantified using a Qubit Fluorometer (ThermoFisher Scientific Inc.). To debranch the amplified mtDNA, 500 ng was treated with 2 µL of T7 endonuclease I, 4 µL of 10X Buffer, and adjusted to a final volume of 40 µL with nuclease-free water. The reaction mixture was then incubated at 37°C for 2 h (New England BioLabs Inc.). The debranched mtDNA was quality-checked using the Genomic DNA ScreenTape assay with the TapeStation system (Agilent Technologies). Before proceeding with any experiments, the DNA was purified using QIAseq Beads (QIAGEN, Germany), following the manufacturer's instructions precisely.

### ONT library preparation and sequencing

During the library preparation process, the Native Barcoding Kit 24 V14 (SQK-NB114) protocol, endorsed by the Nanopore Community (https://nanoporetech.com), was employed to optimize the capability of multiplexing long reads. The sequencing technique using ONT was executed using a Flongle flow cell (R10.4.1). The library preparation protocol included repairing and end-prepping/dA-tailing samples with the NEBNext End Repair/dA-tailing module. A separate

dT-tailed barcode adapter was then added to the dA-tailed templates. Subsequently, the barcoded samples were combined into a single pool. Each barcode adapter had a cohesive end that was connected to the provided sequencing adapter. Finally, the sequencing mixture was loaded into the flow cell. This sequencing run was set up on the MinION instrument for an average of 20 h.

### Pre-processing of mitochondrial genome variation analysis

The raw data was generated by the MinKNOW software (Oxford Nanopore Technologies). During real-time acquisition, the Dorado base-calling method was used in super-accuracy mode to call this data. All FASTQ files associated with each barcode were combined into a single FASTQ file for each sample_code_replicate. The FASTQ files received a first quality check with NanoPlot [13]. Low-quality reads (those with a quality score of less than 9) were eliminated using Nanofilt [14].

We performed the Minimap2 [15] alignment by aligning them to the mtDNA reference genome, revised the Cambridge Reference Sequence (rCRS), and generated aligned reads in SAM format. All SAM files were converted to BAM files and sorted with tools from the Samtools repository [16]. The alignment statistics were derived by estimating the reads that mapped entirely to the mitochondrial sequence using Qualimap [17].

### Mitochondrial genome variations analysis

We utilized mtDNA-Server 2 to detect mitochondrial genome variations with the default parameters. This enhanced platform is accessible through the mitoverse cloud at https://mitoverse.i-med.ac.at [18]. To detect SVs, we utilized Sniffles2 to calls SVs in long-read sequencing data [19]. We investigated depth coverage cutoffs 50x. We used the Ensembl browser to annotate each VCF file to determine the start and end points of each SV [20]. Subsequently, circos plots were generated using ShinyCircos-V2.0, a graphic user interface visualization tool accessible online at https://venyao.xyz/shinyCircos/ [21]. Finally, we reviewed previous studies on the MITOMAP database [22].

## Results

### Defining mitochondrial genome diversity: INDELs, SNVs and SV analysis

The mitoverse results encompass a QC report, haplogroup analysis, and mean depth coverage (S14 Table in S1 File) along with the annotated variants. The number of mtDNA SNVs, insertions, and deletions identified in each cell line showed no significant differences among the cell lines (Fig 1A). However, the number of INDELs within each gene varied, with the D-loop, *MT-RNR2*, *MT-CO1*, *MT-ND4*, and *MT-ND5* genes exhibiting a high frequency of INDELs across all cell lines (Fig 1C and S6–S11 Tables in S1 File). The number of SNVs per gene revealed that the D-loop, *MT-CO1*, *MT-ND4*, *MT-ND5*, and *MT-CYB* genes had a higher number of SNVs in CCA cell lines compared to MMNK-1, particularly in KKU-213A and KKU-452. Interestingly, five CCA cell lines had SNVs in the *MT-ND6* gene, while the MMNK-1 cell line had none (Fig 1D).

To further understand the landscape of mtDNA SVs in CCA cell lines, we performed a comprehensive analysis of deletions, duplications, and inversions using Sniffles2, applying a 50x depth coverage cutoff. The analysis revealed significant variation in the number of SVs among the different cell lines, identifying 208 SVs in KKU-023, 185 SVs in KKU-055, 231 SVs in KKU-100, 69 SVs in KKU-213A, 172 SVs in KKU-452, and 217 SVs in the non-tumorigenic cholangiocyte cell line MMNK-1 (Fig 1B). This variation underscores the genetic diversity within the mtDNA of these cell lines.

### KKU-023 cell line

In the KKU-023 cell line, 48 SNVs were identified, including 14 non-synonymous mutations. Notable mutations such as m.3571C > T, m.3916G > A, m.8462T > C, m.9712T > C, m.14249G > A, m.14772C > T, m.15506G > A, and m.7090G > C exhibited high pathogenicity based on MutPred scores (>0.5), Selection scores (>0.5), and Conservation Index (CI) values

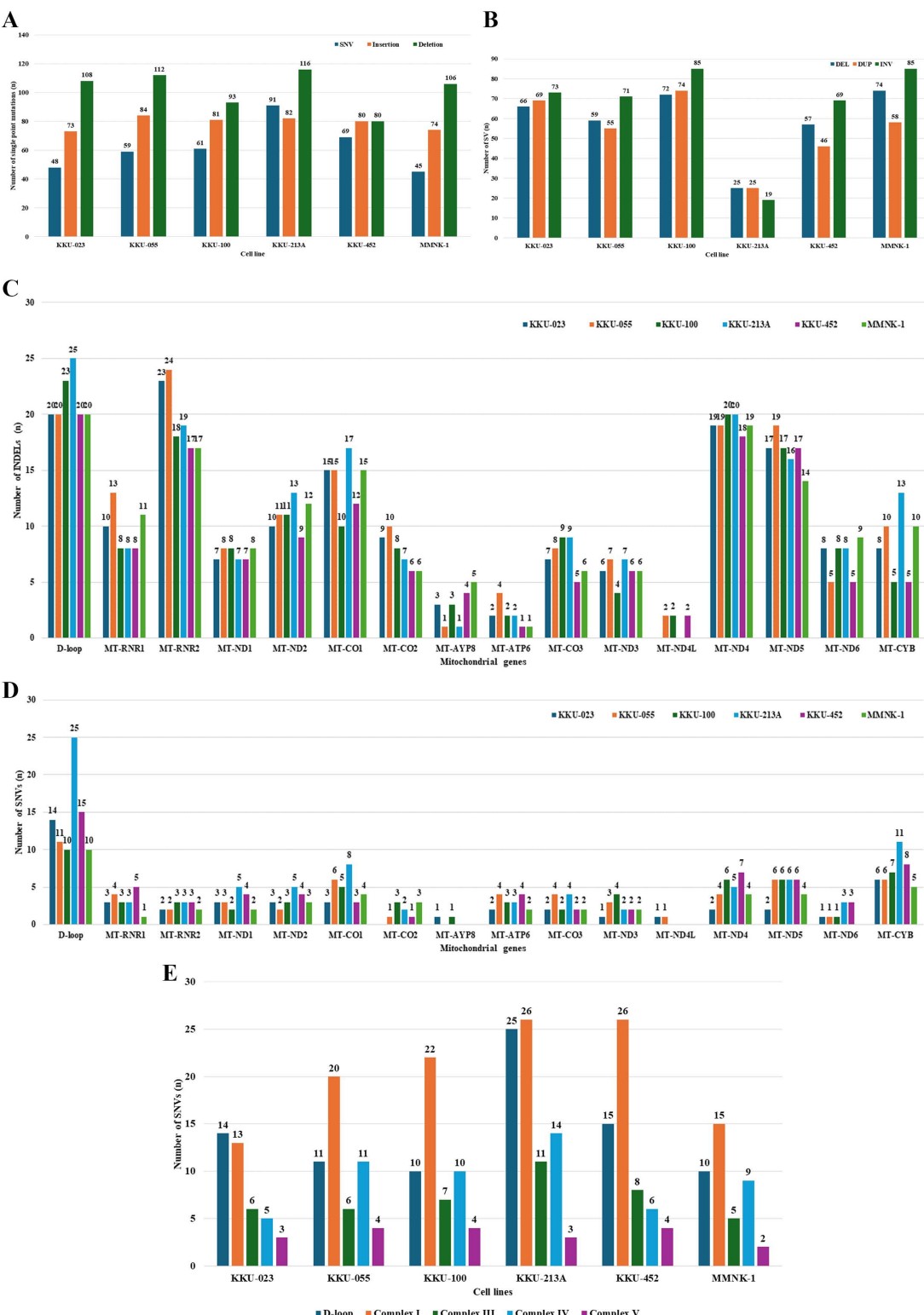

**Fig 1. The number of mtDNA alterations of each cell line.** Number of mtDNA SNVs and INDELs (A) mtDNA structural variant types (B) INDELs in each mitochondrial gene (C) SNVs in each mitochondrial gene and (D) SNVs in each OXPHOS Complex.

**Table 1. SNVs of KKU-023 cell line.**

| SNV | Hetero-plasmy level (%) | Maplocus | Amino Acid (Ref) | New Amino Acid (Variant) | Mut-Pred Score | mtDNA Selection Score | CI MitoTool | OXPHOS complex | MitoMap report and previuos study |
|---|---|---|---|---|---|---|---|---|---|
| m.183A>G | 96.6 | MT-DLOOP2 | – | – | – | – | – | – | POLG/PEO muscle |
| m.203G>A | 2.3 | MT-DLOOP2 | – | – | – | – | – | – | Breast tumor |
| m.204T>C | 97.5 | MT-DLOOP2 | – | – | – | – | – | – | Various tumors, POLG |
| m.263A>G | 100 | MT-DLOOP2 | – | – | – | – | – | – | POLG/MNGIE muscle, CCA and HCC cell lines |
| m.309C>T | 2.4 | MT-DLOOP2 | – | – | – | – | – | – | POLG/PEO muscle |
| m.310T>C | 4.8 | MT-DLOOP2 | – | – | – | – | – | – | |
| m.489T>C | 3.2 | MT-DLOOP2 | – | – | – | – | – | – | Ovarian carcinoma, prostate tumor and CCA cell lines |
| m.750A>G | 99.6 | MT-RNR1 | – | – | – | – | – | – | CCA cell lines |
| m.1438A>G | 98.1 | MT-RNR1 | – | – | – | – | – | – | CCA and HCC cell lines |
| m.1541T>C | 97 | MT-RNR1 | – | – | – | – | – | – | |
| m.2706A>G | 99.5 | MT-RNR2 | – | – | – | – | – | – | CCA and HCC cell lines |
| m.3571C>T | 1.5 | MT-ND1 | Leu | Phe | 0.625 | 0.768 | 0.942 | I | |
| m.3916G>A | 2 | MT-ND1 | Glu | Lys | 0.803 | 1.524 | 1 | I | |
| m.3970C>T | 97.6 | MT-ND1 | – | – | – | – | – | I | |
| m.4769A>G | 99.9 | MT-ND2 | – | – | – | – | – | I | CCA and HCC cell lines |
| m.5442T>C | 1.5 | MT-ND2 | Phe | Leu | 0.437 | 0.372 | 0.231 | I | |
| m.5492T>C | 1.6 | MT-ND2 | – | – | – | – | – | I | |
| m.6620T>C | 97.4 | MT-CO1 | – | – | – | – | – | IV | |
| m.7028C>T | 99.6 | MT-CO1 | – | – | – | – | – | IV | CCA and HCC cell lines |
| m.8462T>C | 98.3 | MT-ATP8 | Tyr | His | 0.718 | 1.099 | 0.885 | V | |
| m.8860A>G | 99.1 | MT-ATP6 | Thr | Ala | 0.369 | 0.287 | 0.750 | V | CCA and HCC cell lines |
| m.9540T>C | 2 | MT-CO3 | – | – | – | – | – | IV | OPA1, CCA and HCC cell lines |
| m.9712T>C | 1.8 | MT-CO3 | Leu | Pro | 0.73 | 1.151 | 1 | IV | |
| m.10398A>G | 2 | MT-ND3 | Thr | Ala | 0.17 | 0.134 | 0.366 | I | Various tumors, CCA and HCC cell lines |
| m.10456A>G | 4.1 | MT-TR | – | – | – | – | – | – | |
| m.10736C>T | 98 | MT-ND4L | – | – | – | – | – | I | |
| m.11611G>A | 1.6 | MT-ND4 | – | – | – | – | – | I | |
| m.11719G>A | 100 | MT-ND4 | – | – | – | – | – | I | CCA and HCC cell lines |
| m.12714T>C | 98.3 | MT-ND5 | – | – | – | – | – | I | |
| m.14249G>A | 1.7 | MT-ND6 | Ala | Val | 0.679 | 0.946 | 0.808 | I | |
| m.14766C>T | 99.9 | MT-CYB | Thr | Ile | 0.165 | 0.131 | 0.692 | III | CCA and HCC cell lines |
| m.14772C>T | 2.5 | MT-CYB | Pro | Leu | 0.74 | 1.196 | 0.904 | III | |
| m.15301G>A | 1.6 | MT-CYB | – | – | – | – | – | III | CCA and HCC cell lines |
| m.15326A>G | 99.9 | MT-CYB | Thr | Ala | 0.452 | 0.395 | 0.712 | III | CCA and HCC cell lines |
| m.15506G>A | 1.5 | MT-CYB | Asp | Asn | 0.596 | 0.687 | 0.981 | III | |
| m.15547C>T | 97.4 | MT-CYB | – | – | – | – | – | III | |
| m.16192C>T | 94.7 | MT-DLOOP1 | – | – | – | – | – | – | Colonic mucosa |
| m.16223C>T | 2.3 | MT-DLOOP1 | – | – | – | – | – | – | Colonic mucosa, tumors, CCA and HCC cell lines |
| m.16304T>C | 97.9 | MT-DLOOP1 | – | – | – | – | – | – | Esophageal, breast and prostate tumors |
| m.16309A>G | 97.1 | MT-DLOOP1 | – | – | – | – | – | – | |
| m.16390G>A | 96.8 | MT-DLOOP1 | – | – | – | – | – | – | Breast and ovarian tumor |
| m.16449C>T | 2 | MT-DLOOP1 | – | – | – | – | – | – | |

*(Continued)*

**Table 1.** (Continued)

| SNV | Hetero-plasmy level (%) | Maplocus | Amino Acid (Ref) | New Amino Acid (Variant) | Mut-Pred Score | mtDNA Selection Score | CI MitoTool | OXPHOS complex | MitoMap report and previuos study |
|---|---|---|---|---|---|---|---|---|---|
| m.16519T>C | 99.4 | MT-DLOOP1 | – | – | – | – | – | – | Glioblastoma, gastric, lung, ovarian, prostate tumors, CCA and HCC cell lines |
| m.1787G>T | 3 | *MT-RNR2* | – | – | – | – | – | – | |
| m.4334A>C | 2.2 | *MT-TQ* | – | – | – | – | – | – | |
| m.7090G>C | 1.3 | *MT-CO1* | Trp | Ser | 0.797 | 1.489 | 1 | IV | |
| m.8592G>C | 1.8 | *MT-ATP6* | – | – | – | – | – | V | |
| m.13928G>C | 97.7 | *MT-ND5* | Ser | Thr | 0.321 | 0.239 | 0.077 | I | |

(>0.8). Five non-synonymous SNVs—m.8462T>C, m.8860A>G, m.14766C>T, m.15326A>G, and m.13928G>C—displayed high heteroplasmy levels of 98.3%, 99.1%, 99.9%, 99.9%, and 97.7%, respectively (Table 1). Most SNVs occurred in the *MT-CYB* and *MT-ND1* genes, located in Complex III and I, respectively. SV analysis revealed 66 deletions, 69 duplications, and 73 inversions (Fig 2A).

**KKU-055 cell line**

The KKU-055 cell line presented 59 SNVs, including 16 non-synonymous mutations. Significant mutations such as m.6667C>T, m.9493G>A, m.9712T>C, m.13406G>A, and m.14249G>A were observed, all demonstrating high scores in MutPred, Selection, and CI. Mutations like m.8860A>G, m.9053G>A, m.10609T>C, m.12406G>A, m.13759G>A, m.15326A>G, and m.13928G>C had heteroplasmy levels exceeding 96% (Table 2). Most SNVs occurred in the *MT-ND5* and *MT-ATP6* genes, located in Complex I and V, respectively, indicating these may be specific pathogenic genes of this poorly differentiated cell line. The SVs in KKU-055 included 59 deletions, 55 duplications, and 71 inversions (Fig 2B).

**KKU-100 cell line**

In the KKU-100 cell line, 61 SNVs were detected, with 16 non-synonymous mutations. Noteworthy mutations such as m.3890G>A, m.4818G>A, m.6762G>A, m.10797C>T, m.11234C>T, m.13633G>A, m.13804G>A, m.15128T>C, m.9172C>A, and m.10095C>A were found, each having high MutPred, Selection, and CI scores. Additionally, mutations m.7775G>A, m.8701A>G, m.8860A>G, m.10398A>G, m.14766C>T, and m.15326A>G showed heteroplasmy levels above 98% (Table 3). Most SNVs occurred in the *MT-CYB* and *MT-ATP6* genes, located in Complex III and V, respectively. This cell line had 72 deletions, 74 duplications, and 85 inversions (Fig 2C).

**KKU-213A cell line**

The KKU-213A cell line exhibited the highest number of SNVs, with 91 identified, including 21 non-synonymous mutations. Key mutations such as m.3571C>T, m.5464T>C, m.8584G>A, m.9529C>T, m.9712T>C, m.14291T>C, m.14904T>C, m.15005G>A, m.15056T>C, m.9537C>A, and m.15024G>C were notable for their high scores in MutPred, Selection, and CI. Mutations m.7853G>A, m.8584G>A, m.8701A>G, m.8860A>G, m.10398A>G, m.12541G>A, m.14291T>C, m.14318T>C, m.14766C>T, m.14904T>C, m.15326A>G, and m.15024G>C exhibited heteroplasmy levels above 92% (Table 4). Most SNVs occurred in the *MT-CYB*, *MT-ATP6*, and *MT-CO3* genes, located in Complex III, V, and IV, respectively. SV analysis revealed 25 deletions, 25 duplications, and 19 inversions (Fig 2D).

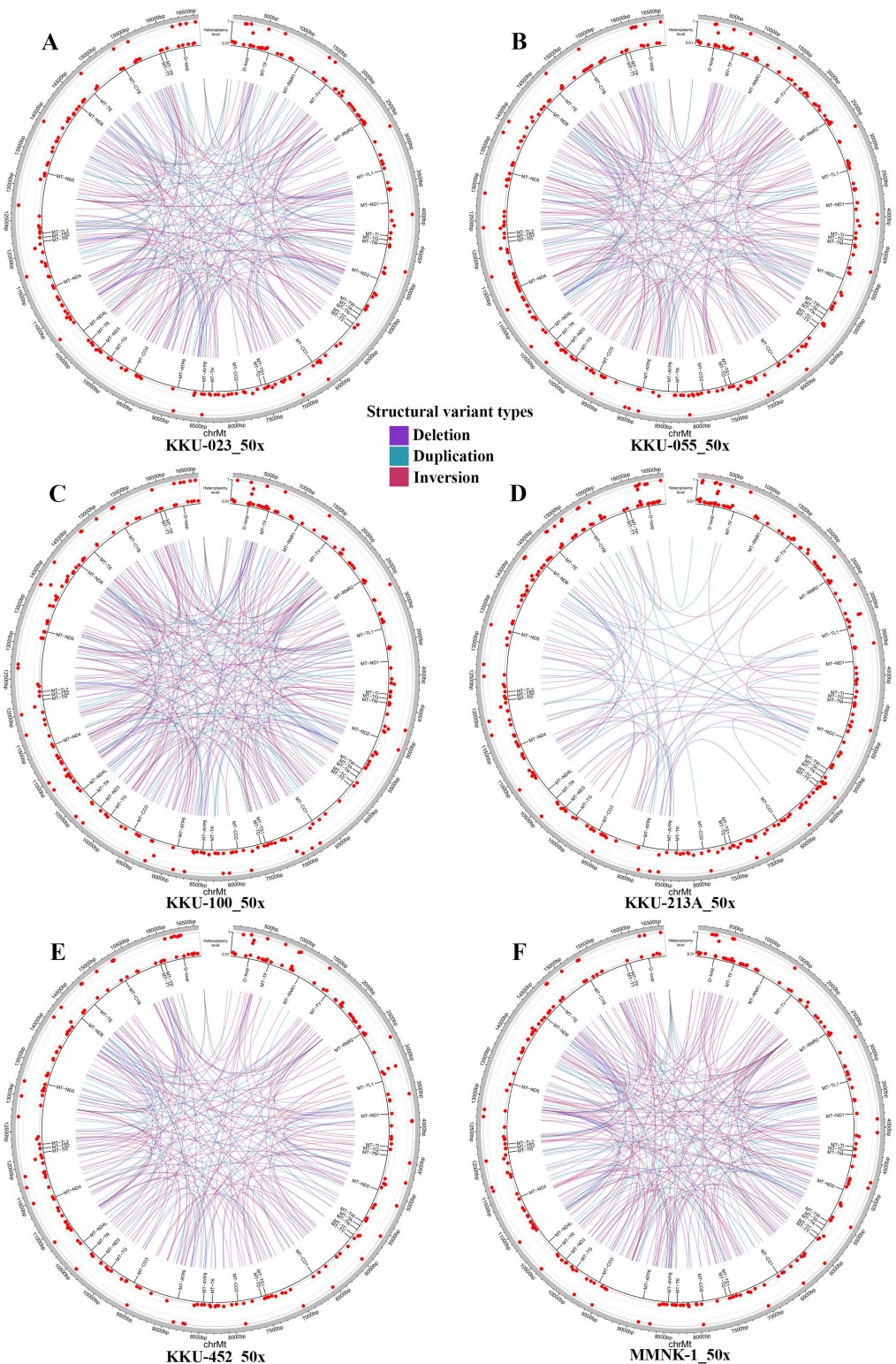

**Fig 2. The circos plots of each cell line.** The circos plots depict the positions of SNVs and INDELs, marked by red dots along the circumference. Structural deletions are indicated by violet lines, structural duplications by blue-green lines, and structural inversions by red-wine lines.

**Table 2.** SNVs of KKU-055 cell line.

| SNV | Hetero-plasmy level (%) | Maplocus | Amino Acid (Ref) | New Amino Acid (Variant) | Mut-Pred Score | mtDNA Selection Score | CI MitoTool | OXPHOS complex | MitoMap report and previous study |
|---|---|---|---|---|---|---|---|---|---|
| m.199T>C | 2.1 | MT-DLOOP2 | – | – | – | – | – | – | Ovarian carcinoma, POLG/MNGIE muscle, and CCA and HCC cell lines |
| m.263A>G | 99.8 | MT-DLOOP2 | – | – | – | – | – | – | POLG/MNGIE muscle, CCA and HCC cell lines |
| m.310T>C | 6.7 | MT-DLOOP2 | – | – | – | – | – | – | |
| m.489T>C | 3.3 | MT-DLOOP2 | – | – | – | – | – | – | Ovarian carcinoma, prostate tumor, and CCA cell lines |
| m.750A>G | 99.9 | MT-RNR1 | – | – | – | – | – | – | CCA and HCC cell lines |
| m.960C>T | 1.7 | MT-RNR1 | – | – | – | – | – | – | |
| m.1187T>C | 96 | MT-RNR1 | – | – | – | – | – | – | |
| m.1438A>G | 98.8 | MT-RNR1 | – | – | – | – | – | – | CCA and HCC cell lines |
| m.2706A>G | 99.7 | MT-RNR2 | – | – | – | – | – | – | CCA and HCC cell lines |
| m.3882G>A | 1.4 | MT-ND1 | – | – | – | – | – | I | |
| m.3970C>T | 97.4 | MT-ND1 | – | – | – | – | – | I | |
| m.4086C>T | 95.8 | MT-ND1 | – | – | – | – | – | I | |
| m.4769A>G | 99.9 | MT-ND2 | – | – | – | – | – | I | CCA and HCC cell lines |
| m.5442T>C | 1 | MT-ND2 | Phe | Leu | 0.437 | 0.373 | 0.231 | I | |
| m.6053C>T | 1.7 | MT-CO1 | – | – | – | – | – | IV | |
| m.6392T>C | 96.8 | MT-CO1 | – | – | – | – | – | IV | |
| m.6455C>T | 2.3 | MT-CO1 | – | – | – | – | – | IV | CCA and HCC cell lines |
| m.6667C>T | 1.9 | MT-CO1 | Ser | Phe | 0.818 | 1.614 | 0.981 | IV | |
| m.6962G>A | 95.8 | MT-CO1 | – | – | – | – | – | IV | |
| m.7028C>T | 99.8 | MT-CO1 | – | – | – | – | – | IV | CCA and HCC cell lines |
| m.8149A>G | 96.4 | MT-CO2 | – | – | – | – | – | IV | |
| m.8701A>G | 3 | MT-ATP6 | Thr | Ala | 0.14 | 0.119 | 0.673 | V | Thyroid tumors, CCA and HCC cell lines |
| m.8860A>G | 98.9 | MT-ATP6 | Thr | Ala | 0.369 | 0.287 | 0.75 | V | CCA and HCC cell lines |
| m.8994G>A | 97.3 | MT-ATP6 | – | – | – | – | – | V | |
| m.9053G>A | 96.6 | MT-ATP6 | Ser | Asn | 0.37 | 0.288 | 0.596 | V | |
| m.9493G>A | 5.8 | MT-CO3 | Gly | Glu | 0.688 | 0.979 | 1 | IV | |
| m.9540T>C | 2.1 | MT-CO3 | – | – | – | – | – | IV | OPA1 and control samples, CCA and HCC cell lines |
| m.9548G>A | 96.2 | MT-CO3 | – | – | – | – | – | IV | POLG and OPA1 samples |
| m.9712T>C | 1.7 | MT-CO3 | Leu | Pro | 0.73 | 1.151 | 1 | IV | |
| m.10310G>A | 97 | MT-ND3 | – | – | – | – | – | I | |
| m.10398A>G | 2.9 | MT-ND3 | Thr | Ala | 0.17 | 0.134 | 0.365 | I | Various tumors, CCA and HCC cell lines |
| m.10400C>T | 3.2 | MT-ND3 | – | – | – | – | – | I | CCA and HCC cell lines |
| m.10609T>C | 96.6 | MT-ND4L | Met | Thr | 0.481 | 0.442 | 0.712 | I | CCA tumor |
| m.11215C>T | 91.2 | MT-ND4 | – | – | – | – | – | I | |
| m.11611G>A | 1.6 | MT-ND4 | – | – | – | – | – | I | |
| m.11665C>T | 1.6 | MT-ND4 | – | – | – | – | – | I | |
| m.11719G>A | 99.9 | MT-ND4 | – | – | – | – | – | I | CCA and HCC cell lines |
| m.12406G>A | 97.8 | MT-ND5 | Val | Ile | 0.251 | 0.182 | 0.135 | I | |
| m.12882C>T | 97.4 | MT-ND5 | – | – | – | – | – | I | |

*(Continued)*

**Table 2.** (Continued)

| SNV | Hetero-plasmy level (%) | Maplocus | Amino Acid (Ref) | New Amino Acid (Variant) | Mut-Pred Score | mtDNA Selection Score | CI MitoTool | OXPHOS complex | MitoMap report and previous study |
|---|---|---|---|---|---|---|---|---|---|
| m.13132C>T | 2.6 | *MT-ND5* | – | – | – | – | – | I | |
| m.13406G>A | 2.1 | *MT-ND5* | Arg | Aln | 0.76 | 1.292 | 1 | I | |
| m.13759G>A | 96.8 | *MT-ND5* | Ala | Thr | 0.251 | 0.182 | 0.0769 | I | |
| m.14249G>A | 1.5 | *MT-ND6* | Ala | Val | 0.679 | 0.946 | 0.808 | I | |
| m.14766C>T | 100 | *MT-CYB* | Thr | Ile | 0.165 | 0.131 | 0.692 | III | CCA and HCC cell lines |
| m.14783T>C | 2.5 | *MT-CYB* | – | – | – | – | – | III | CCA and HCC cell lines |
| m.15043G>A | 3.3 | *MT-CYB* | – | – | – | – | – | III | CCA and HCC cell lines |
| m.15301G>A | 2.6 | *MT-CYB* | – | – | – | – | – | III | CCA and HCC cell lines |
| m.15326A>G | 98.9 | *MT-CYB* | Thr | Ala | 0.452 | 0.395 | 0.712 | III | CCA and HCC cell lines |
| m.16108C>T | 95.4 | MT-DLOOP1 | – | – | – | – | – | – | |
| m.16129G>A | 97.2 | MT-DLOOP1 | – | – | – | – | – | – | CCA and HCC cell lines |
| m.16162A>G | 94.7 | MT-DLOOP1 | – | – | – | – | – | – | |
| m.16172T>C | 96 | MT-DLOOP1 | – | – | – | – | – | – | MNGIE tissues, colonic mucosa, head/neck tumor back-mutation |
| m.16223C>T | 3.3 | MT-DLOOP1 | – | – | – | – | – | – | Colonic mucosa, tumors, CCA and HCC cell lines |
| m.16304T>C | 97.4 | MT-DLOOP1 | – | – | – | – | – | – | Esophageal, breast and prostate tumors |

## KKU-452 cell line

KKU-452 showed 69 SNVs, with 17 being non-synonymous mutations. Significant mutations included m.5464T>C, m.8725A>G, m.9026G>A, m.9712T>C, m.12994G>A, m.14178T>C, m.15005G>A, m.15557G>A, and m.11873A>T, all with high MutPred, Selection, and CI scores. Moreover, mutations m.4491G>A, m.8701A>G, m.8725A>G, m.8860A>G, m.10398A>G, m.12994G>A, m.13204G>A, m.14178T>C, m.14766C>T, m.15317G>A, and m.15326A>G had heteroplasmy levels over 95% (Table 5). Most SNVs occurred in the *MT-CYB* and *MT-ATP6* genes, located in Complex III and V, respectively. The SV analysis revealed 57 deletions, 46 duplications, and 69 inversions (Fig 2E). The identified mtDNA genes and SVs suggest that KKU-452 might have a similar mutation pattern to KKU-100.

## MMNK-1 cell line

The MMNK-1 cell line had 46 SNVs, with 9 non-synonymous mutations. Among these, mutations m.8160A>G, m.13406G>A, and m.13970G>A showed high MutPred, Selection, and CI scores. Additionally, mutations m.5442T>C, m.8701A>G, m.8860A>G, m.10398A>G, m.14766C>T, and m.15326A>G had heteroplasmy levels exceeding 95% (Table 6). There was no high mutation concentration in any specific gene. For SVs, there were 74 deletions, 58 duplications, and 85 inversions (Fig 2F). These mutation patterns highlight the distinct mtDNA mutational landscape observed in the CCA cell lines.

In all cell lines, we identified one shared SNV and seven INDELs across five CCA cell lines (Table 7). Most mutations were concentrated in Complex I genes and the D-loop region. Non-synonymous SNVs ranged from 9 in MMNK-1–21 in KKU-213A, with common SNVs such asm.1787G>T and m.4334A>C found in all six cell lines, noted for their potential pathogenicity and conservation across species. High heteroplasmy levels in mutations like m.8860A>G, m.8701A>G, and m.15326A>G suggest these mutations play a key role in mitochondrial dysfunction in CCA. Additionally, the SNV m.750A>G and seven common INDELs were present in CCA cell lines while high heteroplasmy was also observed in

**Table 3. SNVs of KKU-100 cell line.**

| SNV | Hetero-plasmy level (%) | Maplocus | Amino Acid (Ref) | New Amino Acid (Variant) | Mut-Pred Score | mtDNA Selection Score | CI MitoTool | OXPHOS complex | MitoMap report and previous study |
|---|---|---|---|---|---|---|---|---|---|
| m.73A>G | 99.4 | MT-DLOOP2 | – | – | – | – | – | – | Aging brains, POLG/PEO, thyroid, prostate tumors, CCA and HCC cell lines |
| m.263A>G | 99.8 | MT-DLOOP2 | – | – | – | – | – | – | POLG,MNGIE muscle, CCA and HCC cell lines |
| m.489T>C | 98.9 | MT-DLOOP2 | – | – | – | – | – | – | Ovarian carcinoma, prostate tumor and CCA cell lines |
| m.709G>A | 1.4 | *MT-RNR1* | – | – | – | – | – | – | Thyroid carcinoma (NI-EFVPTC) |
| m.750A>G | 99.7 | *MT-RNR1* | – | – | – | – | – | – | CCA and HCC cell lines |
| m.1438A>G | 98.3 | *MT-RNR1* | – | – | – | – | – | – | CCA and HCC cell lines |
| m.1906G>A | 1.4 | *MT-RNR2* | – | – | – | – | – | – | |
| m.2706A>G | 99.8 | *MT-RNR2* | – | – | – | – | – | – | CCA and HCC cell lines |
| m.3589C>T | 1.7 | *MT-ND1* | – | – | – | – | – | I | |
| m.3890G>A | 1.2 | *MT-ND1* | Arg | Gln | 0.865 | 1.934 | 1 | I | |
| m.4769A>G | 99.9 | *MT-ND2* | – | – | – | – | – | I | CCA and HCC cell lines |
| m.4818G>A | 24.6 | *MT-ND2* | Glu | Lys | 0.822 | 1.639 | 1 | I | |
| m.5054G>A | 98.5 | *MT-ND2* | – | – | – | – | – | I | |
| m.6755G>A | 96.3 | *MT-CO1* | – | – | – | – | – | IV | |
| m.10268C>T | 98.1 | *MT-ND3* | – | – | – | – | – | I | |
| m.10398A>G | 98.5 | *MT-ND3* | Thr | Ala | 0.17 | 0.134 | 0.365 | I | Various tumors, CCA and HCC cell lines |
| m.10400C>T | 98.7 | *MT-ND3* | – | – | – | – | – | I | CCA and HCC cell lines |
| m.10797C>T | 2.1 | *MT-ND4* | Pro | Leu | 0.702 | 1.033 | 1 | I | |
| m.10873T>C | 96.7 | *MT-ND4* | – | – | – | – | – | I | CCA cell lines |
| m.10993G>A | 98.2 | *MT-ND4* | – | – | – | – | – | I | |
| m.11234C>T | 1.7 | *MT-ND4* | Pro | Ser | 0.589 | 0.669 | 1 | I | |
| m.11719G>A | 99.9 | *MT-ND4* | – | – | – | – | – | I | CCA and HCC cell lines |
| m.12007G>A | 97.8 | *MT-ND4* | – | – | – | – | – | I | |
| m.12651G>A | 98 | *MT-ND5* | – | – | – | – | – | I | |
| m.12705C>T | 98.6 | *MT-ND5* | – | – | – | – | – | I | Prostate tumor, CCA and HCC cell lines |
| m.13633G>A | 19.7 | *MT-ND5* | Gly | Ser | 0.714 | 1.082 | 0.827 | I | |
| m.13804G>A | 13.4 | *MT-ND5* | Ala | Thr | 0.658 | 0.872 | 1 | I | |
| m.13851C>T | 98.1 | *MT-ND5* | – | – | – | – | – | I | |
| m.13915G>A | 57 | *MT-ND5* | – | – | – | – | – | I | |
| m.14212T>C | 98.2 | *MT-ND6* | – | – | – | – | – | I | |
| m.14766C>T | 99.7 | *MT-CYB* | Thr | Ile | 0.165 | 0.131 | 0.692 | III | CCA and HCC cell lines |
| m.14783T>C | 98.5 | *MT-CYB* | – | – | – | – | – | III | CCA and HCC cell lines |
| m.15043G>A | 98.5 | *MT-CYB* | – | – | – | – | – | III | CCA and HCC cell lines |
| m.15128T>C | 3.7 | *MT-CYB* | Phe | Leu | 0.679 | 0.946 | 1 | III | |
| m.15301G>A | 98.5 | *MT-CYB* | – | – | – | – | – | III | CCA and HCC cell lines |
| m.15326A>G | 99.3 | *MT-CYB* | Thr | Ala | 0.452 | 0.395 | 0.712 | III | CCA and HCC cell lines |
| m.15868C>T | 2.6 | *MT-CYB* | – | – | – | – | – | III | |
| m.15908T>C | 97.6 | *MT-TT* | – | – | – | – | – | – | |
| m.16223C>T | 99 | MT-DLOOP1 | – | – | – | – | – | – | Colonic mucosa, CCA and HCC cell lines |
| m.16311T>C | 97.5 | MT-DLOOP1 | – | – | – | – | – | – | Colonic mucosa and prostate tumor |
| m.16362T>C | 97.1 | MT-DLOOP1 | – | – | – | – | – | – | |

*(Continued)*

**Table 3.** (Continued)

| SNV | Hetero-plasmy level (%) | Maplocus | Amino Acid (Ref) | New Amino Acid (Variant) | Mut-Pred Score | mtDNA Selection Score | CI MitoTool | OXPHOS complex | MitoMap report and previous study |
|---|---|---|---|---|---|---|---|---|---|
| m.16449C>T | 98.5 | MT-DLOOP1 | – | – | – | – | – | – | |
| m.16519T>C | 98.8 | MT-DLOOP1 | – | – | – | – | – | – | Glioblastoma, gastric, lung, ovarian, prostate tumors, CCA and HCC cell lines |
| m.278A>C | 1.9 | MT-DLOOP2 | – | – | – | – | – | – | |
| m.302A>C | 1.7 | MT-DLOOP2 | – | – | – | – | – | – | |
| m.1787G>T | 2.9 | *MT-RNR2* | – | – | – | – | – | – | |
| m.4334A>C | 2.6 | *MT-TQ* | – | – | – | – | – | – | |
| m.8080C>A | 97.8 | *MT-CO2* | – | – | – | – | – | IV | |
| m.9172C>A | 62.5 | *MT-ATP6* | Leu | Ile | 0.571 | 0.624 | 0.962 | V | |
| m.10095C>A | 1.9 | *MT-ND3* | Leu | Met | 0.63 | 0.783 | 0.712 | I | |

specific SNVs (e.g., m.8462T>C in KKU-023, m.9493G>A in KKU-055, m.9172C>A in KKU-100, m.15024G>C in KKU-213A, m.12994G>A in KKU-452, and m.13406G>A in MMNK-1), indicating their potential involvement in carcinogenesis. Table 8 provides clinical characteristics of patient donors and highlights the high pathogenic mitochondrial genes for each cell line. Specifically, KKU-023 has *MT-CYB* and *MT-ND1*, KKU-055 has *MT-ND5* and *MT-ATP6*, KKU-100 and KKU-452 have *MT-CYB* and *MT-ATP6*, while KKU-213A contains *MT-CYB*, *MT-ATP6*, and *MT-CO3* genes. The substantial variation in SVs among the cell lines, ranging from 69 in KKU-213A to 231 in KKU-100, indicates differences in genetic alterations that may reflect distinct mechanisms driving disease progression. This diversity suggests possible therapeutic targets, warranting further research into the functional impact of these SVs and their role in CCA development.

## Discussion

This study explored the entire mitochondrial genome in CCA lines using ONT. We compared the sequence data to the Revised Cambridge Reference Sequence (rCRS) to identify nucleotide changes. Due to the lack of a corresponding non-cancerous cholangiocyte cell line, we utilized the rCRS and the MMNK-1 cell line, which is an immortalized cholangiocyte created *via* SV40T and hTERT transduction [23], and origin from OUMS-21 which derived from the human embryo liver tissue [24].

Our analysis revealed that most SNVs and INDELs occurred in the D-loop region of mtDNA (Figs 1C–1E), which is known for its hypervariable nature and role in controlling replication. The hypervariable (HV) segments of D-loop, particularly nucleotides 16519 (HV-I) and 73 (HV-II), are commonly associated with cancer risk, as seen in salivary gland tumors and other cancers [25]. The m.73A>G variant has been reported in various conditions, including aging brain, polymerase gamma-progressive external ophthalmoplegia (POLG/PEO), thyroid, and prostate tumors [26–31] and was detected in CCA and hepatocellular carcinoma (HCC) cell lines [32]. Although the m.73A>G mutation has been reported to have no significant impact the stability of mtDNA secondary structure, it has been linked to certain cancers. Interestingly, its role in myocardial infarction seems to be beneficial, potentially by affecting mitochondrial function and gene expression [33]. Similarly, the m.16519T>C mutation has been found in glioblastoma, gastric, lung, ovarian, and prostate tumors [31,34–37] and CCA and HCC cell [32], there was a report indicated that the presence of this mitochondrial variant might predispose individuals with knee osteoarthritis to a heightened baseline inflammatory state involving IL6 [38]. The m.301T>TC insertion and m.16223C>T have been reported in multiple tumor types [39], while m.16304T>C has been linked to esophageal and breast tumors [40,41]. These findings are consistent with the notion that alterations in the D-loop region may be involved in the regulation of the mitochondrial genome and could be associated with carcinogenesis and the progression of CCA cell lines.

**Table 4. SNVs of KKU-213A cell line.**

| SNV | Hetero-plasmy level (%) | Maplocus | Amino Acid (Ref) | New Amino Acid (Variant) | Mut-Pred Score | mtDNA Selection Score | CI MitoTool | OXPHOS complex | MitoMap report and previous study |
|---|---|---|---|---|---|---|---|---|---|
| m.66G>A | 89.6 | MT-DLOOP2 | – | – | – | – | – | – | POLG/PEO muscle |
| m.67G>A | 16.6 | MT-DLOOP2 | – | – | – | – | – | – | POLG/PEO muscle |
| m.73A>G | 100 | MT-DLOOP2 | – | – | – | – | – | – | Aging brains, POLG/PEO, thyroid tumors, prostate tumors, CCA and HCC cell lines |
| m.146T>C | 1.7 | MT-DLOOP2 | – | – | – | – | – | – | Elderly fibroblasts, elderly/AD brains, POLG/PEO, various tumors and HCC cell lines |
| m.199T>C | 2.1 | MT-DLOOP2 | – | – | – | – | – | – | Ovarian carcinoma, POLG/MNGIE muscle, CCA and HCC cell lines |
| m.234A>G | 2.3 | MT-DLOOP2 | – | – | – | – | – | – | |
| m.251G>A | 4.2 | MT-DLOOP2 | – | – | – | – | – | – | |
| m.263A>G | 99.6 | MT-DLOOP2 | – | – | – | – | – | – | POLG/MNGIE muscle, CCA and HCC cell lines |
| m.489T>C | 98.6 | MT-DLOOP2 | – | – | – | – | – | – | Ovarian carcinoma, prostate tumor and CCA cell lines |
| m.750A>G | 99.8 | *MT-RNR1* | – | – | – | – | – | – | CCA and HCC cell lines |
| m.1438A>G | 97.8 | *MT-RNR1* | – | – | – | – | – | – | CCA and HCC cell lines |
| m.1440G>A | 1.7 | *MT-RNR1* | – | – | – | – | – | – | |
| m.2214A>G | 1.8 | *MT-RNR2* | – | – | – | – | – | – | |
| m.2706A>G | 99.7 | *MT-RNR2* | – | – | – | – | – | – | CCA and HCC cell lines |
| m.3571C>T | 1.5 | *MT-ND1* | Leu | Phe | 0.625 | 0.768 | 0.942 | I | |
| m.3882G>A | 1.7 | *MT-ND1* | – | – | – | – | – | I | |
| m.3970C>T | 2.4 | *MT-ND1* | – | – | – | – | – | I | |
| m.4086C>T | 1.3 | *MT-ND1* | – | – | – | – | – | I | |
| m.4457C>T | 3.8 | *MT-TM* | – | – | – | – | – | | |
| m.4769A>G | 99.8 | *MT-ND2* | – | – | – | – | – | I | CCA and HCC cell lines |
| m.5054G>A | 2.4 | *MT-ND2* | – | – | – | – | – | I | |
| m.5442T>C | 2.2 | *MT-ND2* | Phe | Leu | 0.437 | 0.373 | 0.231 | I | |
| m.5464T>C | 1.9 | *MT-ND2* | Leu | Pro | 0.714 | 1.082 | 0.885 | I | |
| m.5492T>C | 1.5 | *MT-ND2* | – | – | – | – | – | I | |
| m.5668G>A | 3.9 | *MT-TN* | – | – | – | – | – | – | |
| m.5821G>A | 92.9 | *MT-TC* | – | – | – | – | – | – | |
| m.6053C>T | 2.1 | *MT-CO1* | – | – | – | – | – | IV | |
| m.6338A>G | 93.6 | *MT-CO1* | – | – | – | – | – | IV | |
| m.6455C>T | 2.1 | *MT-CO1* | – | – | – | – | – | IV | CCA and HCC cell lines |
| m.6620T>C | 2 | *MT-CO1* | – | – | – | – | – | IV | |
| m.6755G>A | 2.2 | *MT-CO1* | – | – | – | – | – | IV | |
| m.6884C>T | 2.5 | *MT-CO1* | – | – | – | – | – | IV | |
| m.7028C>T | 99.7 | *MT-CO1* | – | – | – | – | – | IV | CCA and HCC cell lines |
| m.7853G>A | 94.7 | *MT-CO2* | Val | Ile | 0.249 | 0.181 | 0.212 | IV | CCA cell line |
| m.8167T>C | 2.9 | *MT-CO2* | – | – | – | – | – | IV | |
| m.8584G>A | 94.8 | *MT-ATP6* | Ala | Thr | 0.553 | 0.583 | 0.788 | V | |

*(Continued)*

**Table 4.** (Continued)

| SNV | Hetero-plasmy level (%) | Maplocus | Amino Acid (Ref) | New Amino Acid (Variant) | Mut-Pred Score | mtDNA Selection Score | CI MitoTool | OXPHOS complex | MitoMap report and previous study |
|---|---|---|---|---|---|---|---|---|---|
| m.8701A>G | 96.8 | *MT-ATP6* | Thr | Ala | 0.14 | 0.119 | 0.673 | V | Thyroid tumors, CCA and HCC cell lines |
| m.8860A>G | 99.5 | *MT-ATP6* | Thr | Ala | 0.369 | 0.287 | 0.75 | V | CCA and HCC cell lines |
| m.9529C>T | 1.9 | *MT-CO3* | Pro | Leu | 0.766 | 1.322 | 1 | IV | |
| m.9540T>C | 91.1 | *MT-CO3* | – | – | – | – | – | IV | OPA1 and control samples, CCA and HCC cell lines |
| m.9712T>C | 1.4 | *MT-CO3* | Leu | Pro | 0.73 | 1.151 | 1 | IV | |
| m.10398A>G | 97.1 | *MT-ND3* | Thr | Ala | 0.17 | 0.134 | 0.365 | I | Various tumors, CCA and HCC cell lines |
| m.10400C>T | 96.9 | *MT-ND3* | – | – | – | – | – | I | CCA and HCC cell lines |
| m.10873T>C | 95.5 | *MT-ND4* | – | – | – | – | – | I | CCA cell line |
| m.11611G>A | 1.9 | *MT-ND4* | – | – | – | – | – | I | |
| m.11719G>A | 99.8 | *MT-ND4* | – | – | – | – | – | I | CCA and HCC cell lines |
| m.11914G>A | 94.6 | *MT-ND4* | – | – | – | – | – | I | |
| m.12007G>A | 1.6 | *MT-ND4* | – | – | – | – | – | I | |
| m.12541G>A | 94.7 | *MT-ND5* | Ala | Thr | 0.432 | 0.366 | 0.231 | I | |
| m.12705C>T | 97.9 | *MT-ND5* | – | – | – | – | – | I | Prostate tumor, CCA and HCC cell lines |
| m.13263A>G | 94.4 | *MT-ND5* | – | – | – | – | – | I | |
| m.13851C>T | 1.9 | *MT-ND5* | – | – | – | – | – | I | |
| m.13915G>A | 1.5 | *MT-ND5* | – | – | – | – | – | I | |
| m.14212T>C | 1.7 | *MT-ND6* | – | – | – | – | – | I | |
| m.14291T>C | 93.5 | *MT-ND6* | Glu | Gly | 0.606 | 0.714 | 0.885 | I | |
| m.14318T>C | 92.8 | *MT-ND6* | Asn | Ser | 0.437 | 0.373 | 0.808 | I | |
| m.14766C>T | 99.6 | *MT-CYB* | Thr | Ile | 0.165 | 0.131 | 0.692 | III | CCA and HCC cell lines |
| m.14783T>C | 97.7 | *MT-CYB* | – | – | – | – | – | III | CCA and HCC cell lines |
| m.14904T>C | 93 | *MT-CYB* | Met | Thr | 0.646 | 0.833 | 1 | III | |
| m.15005G>A | 1.2 | *MT-CYB* | Ala | Thr | 0.656 | 0.866 | 1 | III | |
| m.15043G>A | 98.1 | *MT-CYB* | – | – | – | – | – | III | CCA and HCC cell lines |
| m.15056T>C | 1.5 | *MT-CYB* | Tyr | His | 0.836 | 1.730 | 1 | III | |
| m.15073C>T | 3.3 | *MT-CYB* | – | – | – | – | – | III | |
| m.15301G>A | 97.3 | *MT-CYB* | – | – | – | – | – | III | CCA and HCC cell lines |
| m.15326A>G | 98.8 | *MT-CYB* | Thr | Ala | 0.452 | 0.395 | 0.711 | III | CCA and HCC cell lines |
| m.15908T>C | 2 | *MT-TT* | – | – | – | – | – | – | |
| m.16129G>A | 2.6 | MT-DLOOP1 | – | – | – | – | – | – | CCA and HCC cell lines |
| m.16167C>T | 92.4 | MT-DLOOP1 | – | – | – | – | – | – | |
| m.16189T>C | 91.7 | MT-DLOOP1 | – | – | – | – | – | – | Prostate tumor |
| m.16223C>T | 97.4 | MT-DLOOP1 | – | – | – | – | – | – | Colonic mucosa, tumors, CCA and HCC cell lines |
| m.16294C>T | 2.6 | MT-DLOOP1 | – | – | – | – | – | – | |
| m.16295C>T | 2.3 | MT-DLOOP1 | – | – | – | – | – | – | |
| m.16298T>C | 91.1 | MT-DLOOP1 | – | – | – | – | – | – | |
| m.16304T>C | 3 | MT-DLOOP1 | – | – | – | – | – | – | Esophageal, breast & prostate tumors |
| m.16327C>T | 91.7 | MT-DLOOP1 | – | – | – | – | – | – | Colonic mucosa |

*(Continued)*

**Table 4.** (Continued)

| SNV | Hetero-plasmy level (%) | Maplocus | Amino Acid (Ref) | New Amino Acid (Variant) | Mut-Pred Score | mtDNA Selection Score | CI MitoTool | OXPHOS complex | MitoMap report and previous study |
|---|---|---|---|---|---|---|---|---|---|
| m.16362T>C | 1.9 | MT-DLOOP1 | – | – | – | – | – | – | |
| m.16390G>A | 2.3 | MT-DLOOP1 | – | – | – | – | – | – | Breast and ovarian tumor |
| m.16449C>T | 2.9 | MT-DLOOP1 | – | – | – | – | – | – | |
| m.16519T>C | 98.5 | MT-DLOOP1 | – | – | – | – | – | – | Glioblastoma, gastric, lung, ovarian, prostate tumors, CCA and HCC cell lines |
| m.302A>C | 1.8 | MT-DLOOP2 | – | – | – | – | – | – | |
| m.1787G>T | 2.4 | MT-RNR2 | – | – | – | – | – | – | |
| m.3552T>A | 94 | MT-ND1 | – | – | – | – | – | I | |
| m.4334A>C | 3.4 | MT-TQ | – | – | – | – | – | – | |
| m.5755C>G | 2.6 | NA | – | – | – | – | – | – | |
| m.7196C>A | 94.5 | MT-CO1 | – | – | – | – | – | IV | |
| m.9537C>A | 1.7 | MT-CO3 | Gln | Lys | 0.709 | 1.062 | 0.481 | IV | |
| m.13928G>C | 2 | MT-ND5 | Ser | Thr | 0.321 | 0.239 | 0.076 | I | |
| m.15024G>C | 92.7 | MT-CYB | Cys | Ser | 0.659 | 0.876 | 1 | III | |
| m.15487A>T | 94.9 | MT-CYB | – | – | – | – | – | III | |
| m.16183A>C | 74.8 | MT-DLOOP1 | – | – | – | – | – | – | Lung tumor back-mutation, prostate tumor and colon tumor |

In the coding regions, the highest number of SNVs were found in the genes encoding Complexes I, IV, III, and V, with counts of 122, 55, 43, and 20, respectively (Fig 1E). Previous studies have noted that Complex I mutations are particularly prevalent in CCA tumors [42]. Complex I, also known as NADH-ubiquinone oxidoreductase, is the largest component of the mitochondrial electron transport chain and provides about 40% of the proton motive force necessary for ATP synthesis. It also plays a crucial role in biosynthesis, redox regulation, cell proliferation, resistance to apoptosis, and metastasis. Mutations in Complex I genes are associated with the progression of various cancers, including prostate, thyroid, breast, lung, renal, colorectal, and head and neck tumors [43]. Maintaining the NAD$^+$/NADH ratio *via* Complex I is critical for adaptive responses to hypoxia, including stabilization of hypoxia-inducible factor 1-alpha (HIF1α) [44], and promoting a metabolic shift towards aerobic glycolysis, known as the Warburg effect [45,46]. Complex IV, or cytochrome c oxidase (COX), is a vital enzyme in the mitochondrial ETC responsible for the final electron transfer to oxygen, producing water [47]. Mutations in Complex IV genes can influence breast cancer progression and may serve as potential genetic markers [48]. Complex III, also known as cytochrome c reductase, transfers electrons from coenzyme Q (CoQ) to cytochrome c and moves four protons into the intermembrane space (IMS) [49]. It plays a key role in cellular signaling, and its modulation can significantly impact gene expression, potentially linking its activity to advanced tumor stages and drug resistance, particularly in highly oxidative tumors [50]. Complex V, known as ATP synthase, is crucial for ATP production. In cancer, Complex V is vital due to the high energy demands of rapidly growing cancer cells [49]. High expression levels of ATP synthase-related proteins have been observed in several cancers, including glioma, ovarian, prostate, breast, and clear cell renal cell carcinoma, correlating with poor prognosis [51]. These underscore the significant presence of SNVs in the mitochondrial genes encoding Complexes I, IV, III, and V in CCA cell lines.

The study also identified varying numbers of non-synonymous SNVs across the CCA cell lines: 14 in KKU-023, 16 in KKU-055, 16 in KKU-100, 21 in KKU-213A, 17 in KKU-452, and 9 in MMNK-1. Each cell line exhibited unique INDELs (S6–S11 Tables in S1 File) and SNV profiles (Tables 2–7), but some SNVs were shared. For instance, the m.8701A>G

**Table 5. SNVs of KKU-452 cell line.**

| SNV | Hetero-plasmy level (%) | Maplocus | Amino Acid (Ref) | New Amino Acid (Variant) | Mut-Pred Score | mtDNA Selection Score | CI MitoTool | OXPHOS complex | MitoMap report and previous study |
|---|---|---|---|---|---|---|---|---|---|
| m.151C>T | 94.8 | MT-DLOOP2 | – | – | – | – | – | – | |
| m.152T>C | 94.5 | MT-DLOOP2 | – | – | – | – | – | – | Elderly brains, elderly fibroblasts, ovarian carcinoma, breast tumor |
| m.234A>G | 1.6 | MT-DLOOP2 | – | – | – | – | – | – | |
| m.263A>G | 99.6 | MT-DLOOP2 | – | – | – | – | – | – | POLG/MNGIE muscle, CCA and HCC cell lines |
| m.489T>C | 98.8 | MT-DLOOP2 | – | – | – | – | – | – | Ovarian carcinoma, prostate tumor and CCA cell lines |
| m.750A>G | 99.5 | *MT-RNR1* | – | – | – | – | – | – | CCA and HCC cell lines |
| m.961T>C | 90.3 | *MT-RNR1* | – | – | – | – | – | – | |
| m.988G>A | 95.8 | *MT-RNR1* | – | – | – | – | – | – | |
| m.1438A>G | 98.7 | *MT-RNR1* | – | – | – | – | – | – | CCA and HCC cell lines |
| m.2214A>G | 2.3 | *MT-RNR2* | – | – | – | – | – | – | |
| m.2706A>G | 99.7 | *MT-RNR2* | – | – | – | – | – | – | CCA and HCC cell lines |
| m.3531G>A | 96.6 | *MT-ND1* | – | – | – | – | – | I | |
| m.3906T>C | 95.9 | *MT-ND1* | – | – | – | – | – | I | |
| m.3915G>A | 94.7 | *MT-ND1* | – | – | – | – | – | I | |
| m.3970C>T | 1.9 | *MT-ND1* | – | – | – | – | – | I | |
| m.4491G>A | 97.5 | *MT-ND2* | Val | Ile | 0.439 | 0.376 | 0.173 | I | CCA cell line |
| m.4769A>G | 99.9 | *MT-ND2* | – | – | – | – | – | I | CCA and HCC cell lines |
| m.5108T>C | 96.1 | *MT-ND2* | – | – | – | – | – | I | |
| m.5464T>C | 1.4 | *MT-ND2* | Leu | Pro | 0.714 | 1.082 | 0.885 | I | |
| m.5581A>G | 95.6 | NA | – | – | – | – | – | – | |
| m.5988C>T | 91.1 | *MT-CO1* | – | – | – | – | – | IV | |
| m.6548C>T | 96.4 | *MT-CO1* | – | – | – | – | – | IV | |
| m.7028C>T | 99.4 | *MT-CO1* | – | – | – | – | – | IV | CCA and HCC cell lines |
| m.7861T>C | 97.5 | *MT-CO2* | – | – | – | – | – | IV | |
| m.8701A>G | 98.7 | *MT-ATP6* | Thr | Ala | 0.14 | 0.119 | 0.673 | V | Thyroid tumors, CCA and HCC cell lines |
| m.8725A>G | 97.2 | *MT-ATP6* | Thr | Ala | 0.559 | 0.596 | 0.923 | V | |
| m.8860A>G | 99.4 | *MT-ATP6* | Thr | Ala | 0.369 | 0.287 | 0.75 | V | CCA and HCC cell lines |
| m.9026G>A | 12.9 | *MT-ATP6* | Gly | Asp | 0.869 | 1.964 | 1 | V | |
| m.9540T>C | 97.1 | *MT-CO3* | – | – | – | – | – | IV | OPA1 and control samples, CCA and HCC cell lines |
| m.9712T>C | 1.7 | *MT-CO3* | Leu | Pro | 0.73 | 1.151 | 1 | IV | |
| m.10398A>G | 98.4 | *MT-ND3* | Thr | Ala | 0.17 | 0.134 | 0.365 | I | Various tumors, CCA and HCC cell lines |
| m.10400C>T | 98.3 | *MT-ND3* | – | – | – | – | – | I | CCA and HCC cell lines |
| m.10873T>C | 95.1 | *MT-ND4* | – | – | – | – | – | I | CCA cell line |
| m.11482T>C | 95 | *MT-ND4* | – | – | – | – | – | I | |
| m.11719G>A | 99.9 | *MT-ND4* | – | – | – | – | – | I | CCA and HCC cell lines |
| m.11872C>T | 92.4 | *MT-ND4* | – | – | – | – | – | I | |
| m.12007G>A | 1.3 | *MT-ND4* | – | – | – | – | – | I | |
| m.12091T>C | 1.3 | *MT-ND4* | – | – | – | – | – | I | |
| m.12705C>T | 98.7 | *MT-ND5* | – | – | – | – | – | I | Prostate tumor, CCA and HCC cell lines |
| m.12899G>A | 3.4 | *MT-ND5* | – | – | – | – | – | I | |
| m.12994G>A | 95.9 | *MT-ND5* | Ala | Thr | 0.727 | 1.138 | 1 | I | |

*(Continued)*

**Table 5.** (Continued)

| SNV | Hetero-plasmy level (%) | Maplocus | Amino Acid (Ref) | New Amino Acid (Variant) | Mut-Pred Score | mtDNA Selection Score | CI MitoTool | OXPHOS complex | MitoMap report and previous study |
|---|---|---|---|---|---|---|---|---|---|
| m.13132C>T | 1.7 | *MT-ND5* | – | – | – | – | – | I | |
| m.13204G>A | 96.8 | *MT-ND5* | Val | Ile | 0.413 | 0.340 | 0.135 | I | |
| m.13851C>T | 1.1 | *MT-ND5* | – | – | – | – | – | I | |
| m.14178T>C | 95.3 | *MT-ND6* | Ile | Val | 0.516 | 0.505 | 0.788 | I | |
| m.14212T>C | 1.3 | *MT-ND6* | – | – | – | – | – | I | |
| m.14347A>G | 95 | *MT-ND6* | – | – | – | – | – | I | |
| m.14766C>T | 99.8 | *MT-CYB* | Thr | Ile | 0.165 | 0.131 | 0.692 | III | CCA and HCC cell lines |
| m.14783T>C | 98.5 | *MT-CYB* | – | – | – | – | – | III | CCA and HCC cell lines |
| m.15005G>A | 1.4 | *MT-CYB* | Ala | Thr | 0.656 | 0.866 | 1 | III | |
| m.15043G>A | 98.8 | *MT-CYB* | – | – | – | – | – | III | CCA and HCC cell lines |
| m.15301G>A | 98 | *MT-CYB* | – | – | – | – | – | III | CCA and HCC cell lines |
| m.15317G>A | 95.8 | *MT-CYB* | Ala | Thr | 0.311 | 0.230 | 0.904 | III | |
| m.15326A>G | 99.2 | *MT-CYB* | Thr | Ala | 0.452 | 0.395 | 0.712 | III | CCA and HCC cell lines |
| m.15557G>A | 1.1 | *MT-CYB* | Glu | Lys | 0.796 | 1.483 | 1 | III | |
| m.16086T>C | 96.4 | MT-DLOOP1 | – | – | – | – | – | – | |
| m.16179C>T | 96.2 | MT-DLOOP1 | – | – | – | – | – | – | |
| m.16223C>T | 98.1 | MT-DLOOP1 | – | – | – | – | – | – | Colonic mucosa, CCA and HCC cell lines |
| m.16264C>T | 92.9 | MT-DLOOP1 | – | – | – | – | – | – | |
| m.16287C>T | 96 | MT-DLOOP1 | – | – | – | – | – | – | |
| m.16294C>T | 96.1 | MT-DLOOP1 | – | – | – | – | – | – | |
| m.16304T>C | 2 | MT-DLOOP1 | – | – | – | – | – | – | Esophageal, breast and prostate tumors |
| m.16305A>G | 95.7 | MT-DLOOP1 | – | – | – | – | – | – | |
| m.16311T>C | 96.9 | MT-DLOOP1 | – | – | – | – | – | – | Colonic mucosa, prostate tumor |
| m.16519T>C | 3.9 | MT-DLOOP1 | – | – | – | – | – | – | Glioblastoma, gastric, lung, ovarian, prostate tumors, CCA and HCC cell lines |
| m.955A>C | 6.1 | *MT-RNR1* | – | – | – | – | – | – | |
| m.1787G>T | 4.5 | *MT-RNR2* | – | – | – | – | – | – | |
| m.4334A>C | 2.6 | *MT-TQ* | – | – | – | – | – | – | |
| m.11873A>T | 2.3 | *MT-ND4* | Thr | Ser | 0.586 | 0.661 | 0.731 | I | |

mutation, present in four CCA cell lines (except KKU-023) and MMNK-1, has been reported in thyroid tumors [28,39,49]. The m.10398A>G mutation was found in all six cell lines and has been reported in various cancers [31,49,52,53]. Both SNVs have been observed in CCA and HCC cell lines [32]. Additionally, the m.10609T>C mutation was found in CCA tumors and is located in the MT-ND4L gene, resulting in an amino acid change from isoleucine (I) to threonine (T) [42]. Additionally, in the KKU-213A CCA cell line, which harbors numerous mtDNA SNVs, prior studies have shown that ALDH1A3 expression significantly rises under lactic acidosis (LA) conditions and correlates with LDHA expression. Higher ALDH1A3 levels have been linked to poorer patient survival, indicating its potential as a prognostic marker. The EGFR pathway has been identified as a main regulator of ALDH1A3 in LA, promoting tumor aggressiveness and resistance to gemcitabine. Therefore, targeting the EGFR-ALDH1A3 axis could be a promising therapeutic approach for metastatic CCA in this cell line [54]. The doubling time and the total number of mtDNA SNVs and INDELs for each cell line (S15 Table and S6 Fig in S1 File) illustrating the negative relationship, which suggested a potential link to proliferative capacity. However, this correlation was not statistically significant. Additionally, the expression level of IL-6 and CK19 has been

**Table 6.  SNVs of MMNK-1 cell line.**

| SNV | Hetero-plasmy level (%) | Maplocus | Amino Acid (Ref) | New Amino Acid (Variant) | Mut-Pred Score | mtDNA Selection Score | CI MitoTool | OXPHOS complex | MitoMap report and previous study |
|---|---|---|---|---|---|---|---|---|---|
| m.199T>C | 98 | MT-DLOOP2 | – | – | – | – | – | – | Ovarian carcinoma and POLG/MNGIE muscle |
| m.234A>G | 97.7 | MT-DLOOP2 | – | – | – | – | – | – | |
| m.263A>G | 99.3 | MT-DLOOP2 | – | – | – | – | – | – | POLG/MNGIE muscle, CCA and HCC cell lines |
| m.309C>T | 3.3 | MT-DLOOP2 | – | – | – | – | – | – | POLG/PEO muscle |
| m.310T>C | 4.6 | MT-DLOOP2 | – | – | – | – | – | – | |
| m.489T>C | 99.6 | MT-DLOOP2 | – | – | – | – | – | – | Ovarian carcinoma and prostate tumor |
| m.1438A>G | 99 | *MT-RNR1* | – | – | – | – | – | – | CCA and HCC cell lines |
| m.2706A>G | 99.8 | *MT-RNR2* | – | – | – | – | – | – | CCA and HCC cell lines |
| m.3882G>A | 97.5 | *MT-ND1* | – | – | – | – | – | I | |
| m.4071C>T | 96.9 | *MT-ND1* | – | – | – | – | – | I | CCA and HCC cell lines |
| m.4769A>G | 99.7 | *MT-ND2* | – | – | – | – | – | I | CCA and HCC cell lines |
| m.4850C>T | 96.4 | *MT-ND2* | – | – | – | – | – | I | |
| m.5442T>C | 95.5 | *MT-ND2* | Phe | Leu | 0.437 | 0.373 | 0.231 | I | |
| m.6053C>T | 97.8 | *MT-CO1* | – | – | – | – | – | IV | |
| m.6314C>T | 2.1 | *MT-CO1* | – | – | – | – | – | IV | |
| m.6455C>T | 98.3 | *MT-CO1* | – | – | – | – | – | IV | CCA and HCC cell lines |
| m.7028C>T | 99.8 | *MT-CO1* | – | – | – | – | – | IV | CCA and HCC cell lines |
| m.7961T>C | 97.5 | *MT-CO2* | – | – | – | – | – | IV | |
| m.8160A>G | 2.9 | *MT-CO2* | Tyr | Cys | 0.82 | 1.627 | 0.942 | IV | |
| m.8164C>T | 2.3 | *MT-CO2* | – | – | – | – | – | IV | |
| m.8701A>G | 98.5 | *MT-ATP6* | Thr | Ala | 0.14 | 0.119 | 0.673 | V | Thyroid tumors, CCA and HCC cell lines |
| m.8860A>G | 99.3 | *MT-ATP6* | Thr | Ala | 0.369 | 0.287 | 0.75 | V | CCA and HCC cell lines |
| m.9540T>C | 97.2 | *MT-CO3* | – | – | – | – | – | IV | OPA1 and control samples, CCA and HCC cell lines |
| m.9824T>C | 96.4 | *MT-CO3* | – | – | – | – | – | IV | CCA cell line |
| m.10398A>G | 98.4 | *MT-ND3* | Thr | Ala | 0.17 | 0.134 | 0.365 | I | Various tumors, CCA and HCC cell lines |
| m.10400C>T | 98.6 | *MT-ND3* | – | – | – | – | – | I | CCA and HCC cell lines |
| m.10873T>C | 95.6 | *MT-ND4* | – | – | – | – | – | I | |
| m.11665C>T | 97.9 | *MT-ND4* | – | – | – | – | – | I | |
| m.11719G>A | 99.8 | *MT-ND4* | – | – | – | – | – | I | CCA and HCC cell lines |
| m.12091T>C | 97.4 | *MT-ND4* | – | – | – | – | – | I | |
| m.12705C>T | 99.3 | *MT-ND5* | – | – | – | – | – | I | Prostate tumor, CCA and HCC cell lines |
| m.12804T>C | 97.9 | *MT-ND5* | – | – | – | – | – | I | |
| m.13406G>A | 1.8 | *MT-ND5* | Arg | Gln | 0.76 | 1.292 | 1 | I | |
| m.13970G>A | 7.8 | *MT-ND5* | Ser | Asn | 0.747 | 1.229 | 1 | I | |
| m.14766C>T | 99.8 | *MT-CYB* | Thr | Ile | 0.165 | 0.131 | 0.692 | III | CCA and HCC cell lines |
| m.14783T>C | 99.1 | *MT-CYB* | – | – | – | – | – | III | CCA and HCC cell lines |
| m.15043G>A | 99.4 | *MT-CYB* | – | – | – | – | – | III | CCA and HCC cell lines |
| m.15301G>A | 98.9 | *MT-CYB* | – | – | – | – | – | III | CCA and HCC cell lines |
| m.15326A>G | 99 | *MT-CYB* | Thr | Ala | 0.452 | 0.395 | 0.712 | III | CCA and HCC cell lines |
| m.16223C>T | 98.6 | MT-DLOOP1 | – | – | – | – | – | – | Colonic mucosa, CCA and HCC cell lines |
| m.16294C>T | 98.2 | MT-DLOOP1 | – | – | – | – | – | – | |

*(Continued)*

**Table 6.** (Continued)

| SNV | Hetero-plasmy level (%) | Maplocus | Amino Acid (Ref) | New Amino Acid (Variant) | Mut-Pred Score | mtDNA Selection Score | CI MitoTool | OXPHOS complex | MitoMap report and previous study |
|---|---|---|---|---|---|---|---|---|---|
| m.16295C>T | 97.8 | MT-DLOOP1 | – | – | – | – | – | – | |
| m.16519T>C | 99.6 | MT-DLOOP1 | – | – | – | – | – | – | Glioblastoma, gastric, lung, ovarian, prostate tumors, CCA and HCC cell lines |
| m.1787G>T | 2.7 | *MT-RNR2* | – | – | – | – | – | – | |
| m.4334A>C | 2.1 | *MT-TQ* | – | – | – | – | – | – | |

**Table 7. Common alteration among five CCA cell lines.**

| Position | Reference | Variant | Heteroplasmy level (%) | | | | | Substitution/INDEL | Maplocus |
|---|---|---|---|---|---|---|---|---|---|
| | | | KKU-023 | KKU-055 | KKU-100 | KKU-213A | KKU-452 | | |
| 750 | A | G | 99.6 | 99.9 | 99.7 | 99.8 | 99.5 | transition | *MT-RNR1* |
| 2226 | T | TA | 3.2 | 3.2 | 3.3 | 87.9 | 2.3 | insertion | *MT-RNR2* |
| 3167 | TC | T | 4.3 | 8.8 | 7.9 | 9.4 | 9.6 | deletion | *MT-RNR2* |
| 7231 | AC | A | 4.2 | 5 | 4.7 | 6.2 | 6.8 | deletion | *MT-CO1* |
| 7842 | TA | T | 2.7 | 2.3 | 2.8 | 5 | 5 | deletion | *MT-CO2* |
| 11466 | T | TA | 3.4 | 2.3 | 4.9 | 3.3 | 4.3 | insertion | *MT-ND4* |
| 13365 | CG | C | 4.5 | 4.7 | 2.7 | 3.5 | 7.3 | deletion | *MT-ND5* |
| 13370 | C | CT | 4.5 | 2.3 | 2.7 | 4.6 | 4.3 | insertion | *MT-ND5* |

**Table 8. Clinical characteristics of the patient donors and high pathogenic mitochondrial genes of each cell line.**

| Cell line | High pathogenic mitochondrial genes | Clinical data from patient donors | | | | | |
|---|---|---|---|---|---|---|---|
| | | Sex | Age (year-old) | Doubling time (h) | Tumor differentiation/profiles | Metastasis | Morphology |
| KKU-023 | *MT-CYB* *MT-ND1* | F | 64 | 34.8 | Human well differentiated cholangiocarcinoma cell line established from biliary tract. | Yes | Epithelial-like |
| KKU-055 | *MT-ND5* *MT-ATP6* | M | 56 | 24 | Human poorly differentiated cholangiocarcinoma cell line established from biliary tract. | NA | Epithelial-like |
| KKU-100 | *MT-CYB* *MT-ATP6* | F | 65 | 72 | Human poorly differentiated cholangiocarcinoma cell line established from biliary tract. | NA | Epithelial-like |
| KKU-213A | *MT-CYB* *MT-ATP6* *MT-CO3* | M | 58 | 23 | Human mixed (papillary and non-papillary) chol-angiocarcinoma cell line established from biliary tract. | NA | Epithelial-like |
| KKU-452 | *MT-CYB* *MT-ATP6* | F | 63 | 17.9 | Human poorly differentiated cholangiocarcinoma cell line established from biliary tract. | Yes | Epithelial-like |
| MMNK-1 | No trending in specific gene | NA | NA | 40 | Highly differentiated immortalized human hepato-cyte cell line. | NA | Epithelial-like |

investigated using western blot (S7–S8 Fig in S1 File) and droplet digital PCR (S11 Fig in S1 File) analyses. The correlation of their expression with the total number of mtDNA SNVs and INDELs for each cell line was evaluated, however, the analyses did not yield statistically significant results (S9–S10, S12–S13 Fig in S1 File). These findings suggest that further molecular characterization with the larger number of samples is necessary. These findings highlight the presence of potentially pathogenic mutations within mitochondrial complexes and emphasize the substantial accumulation of SNVs in genes encoding key mitochondrial complexes (I, IV, III, and V) of the electron transport chain in CCA cell lines. The high

occurrence of SNVs in the D-loop and coding regions of mtDNA. The unique mutation profiles and shared SNVs across cell lines underscore the relevance of mitochondrial genome alterations as both potential biomarkers and therapeutic targets in CCA.

For the mtDNA SVs in this study can be detected by ONT. The distribution helps highlight the specific types of genetic alterations predominant in each cell line, providing a baseline to assess the impact of these variants in CCA cell lines. The variability observed could be due to differences in the cell lines' origins, genetic backgrounds, or levels of mitochondrial dysfunction. MtDNA deletions and duplications can disrupt mitochondrial gene function, often caused by spontaneous changes or mutations in nuclear-encoded proteins like DNA polymerase γ (POLγ) and Twinkle helicase. These structural alterations are commonly linked to mitochondrial disorders and conditions such as cancer, diabetes, neurodegenerative diseases, and aging. The accumulation of these mtDNA changes in cancer can lead to increased oxidative stress and altered apoptotic pathways, which may promote tumorigenesis [55]. While mtDNA inversions can disrupt mitochondrial genes and may result in the formation of hybrid gene products that impair mitochondrial function and disrupt proteostasis, crucial for cellular health. Inversions caused by inverted repeats (IRs) are suggested to be more mutagenic than deletions caused by direct repeats (DRs). This heightened mutagenicity can lead to greater instability in the mitochondrial genome, increasing the risk of diseases [56]. The incorporation of mtDNA segments into nuclear DNA is often associated with specific processes that lead to structural variations in the nuclear genome.

This highlights the importance of the mitochondrial genome in understanding the complex molecular patterns observed in cancer genomes and in identifying potential cancer-driving events. The positive correlation between the mutation burdens of mitochondrial and nuclear genomes in various cancer types suggests that SVs in mtDNA may influence or be indicative of changes in the nuclear genome [57]. Nevertheless, this is the initial study examining SVs in mtDNA within CCA cell lines. Future research should focus on elucidating the mechanisms by which these mtDNA mutations contribute to mitochondrial dysfunction and CCA progression. Exploring the interactions between mitochondrial and nuclear gene variants could provide deeper insights into the molecular underpinnings of CCA and identify potential therapeutic targets. Speculatively, targeting specific mtDNA mutations or their resulting dysfunctions may offer new avenues for CCA treatment.

This study explored the mtDNA profiles in five CCA cell lines (KKU-023, KKU-055, KKU-100, KKU-213A, KKU-452) and a non-tumorous cholangiocyte cell line (MMNK-1), with a focus on identifying INDELs, SNVs, and SVs. Our key findings reveal a significant presence of SNVs and INDELs, particularly in the D-loop region and genes encoding mitochondrial respiratory complexes, with Complex I mutations being notably prevalent. Additionally, we observed a considerable number of SVs, including deletions, duplications, and inversions, across the different cell lines. Our study contributes to the existing knowledge by highlighting the variability in mtDNA alterations among CCA cell lines and suggesting potential roles these mutations may play in mitochondrial dysfunction and cancer progression. Notably, this is the first study to comprehensively investigate mtDNA SVs in CCA cell lines, providing a baseline for future studies.

## Supporting information

**S1 File. Supporting information.**
(DOCX)

## Author contributions

**Conceptualization:** Arporn Wangwiwatsin, Nisana Namwat, Poramate Klanrit, Luke Boulter, Watcharin Loilome.

**Formal analysis:** Athitaya Faipan, Sirinya Sitthirak, Arporn Wangwiwatsin, Poramate Klanrit.

**Funding acquisition:** Watcharin Loilome.

**Methodology:** Hasaya Dokduang.

**Project administration:** Watcharin Loilome.

**Supervision:** Watcharin Loilome.

**Visualization:** Athitaya Faipan, Sirinya Sitthirak.

**Writing – original draft:** Athitaya Faipan.

**Writing – review & editing:** Sirinya Sitthirak, Arporn Wangwiwatsin, Nisana Namwat, Poramate Klanrit, Attapol Titapun, Apiwat Jareanrat, Vasin Thanasukarn, Natcha Khuntikeo, Luke Boulter, Watcharin Loilome.

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
