## [Decision Letter · Decision Letter 0]

9 Oct 2024

PONE-D-24-41324Mitochondrial genomic alterations in cholangiocarcinoma cell linesPLOS ONE

Dear Dr. Loilome,

Thank you for submitting your manuscript to PLOS ONE. After careful consideration, we feel that it has merit but does not fully meet PLOS ONE’s publication criteria as it currently stands. Therefore, we invite you to submit a revised version of the manuscript that addresses the points raised by the reviewer.

We look forward to receiving your revised manuscript.

Kind regards,

Matias A Avila, Ph.D.

Academic Editor

PLOS ONE

Reviewers' comments:

Reviewer's Responses to Questions

**Comments to the Author**

1. Is the manuscript technically sound, and do the data support the conclusions?

Reviewer #1: Yes

2. Has the statistical analysis been performed appropriately and rigorously? 

Reviewer #1: Yes

3. Have the authors made all data underlying the findings in their manuscript fully available?

Reviewer #1: Yes

4. Is the manuscript presented in an intelligible fashion and written in standard English?

Reviewer #1: Yes

5. Review Comments to the Author

Reviewer #1: The manuscript entitled "Mitochondrial genomic alterations in cholangiocarcinoma cell lines" (PONE-D-24-41324) presents an analysis of mtDNA structural variants in cholangiocarcinoma (CCA)-derived cell lines. While the study provides interesting and valuable insights, the current presentation is predominantly descriptive. To enhance the impact and relevance of the findings, additional experimental validation and functional characterization of the identified structural variants are warranted.

Major Revisions:

1. To establish the functional significance of the observed mtDNA mutations, the authors should evaluate the expression levels of the genes harboring these mutations through qPCR and confirm protein expression via western blotting. These analyses would provide crucial information on whether the identified mutations result in corresponding changes at the transcriptional or translational levels.

2. The manuscript would benefit from the inclusion of phenotypic characterization of the CCA-derived cell lines. Specifically, conducting colony formation assays and determining the doubling time of each cell line would offer important insights into their proliferative and clonogenic capacities, further contributing to understanding their oncogenic potential.

3. It is also recommended to assess the expression of cholangiocarcinoma-related genes by qPCR, including markers such as CK19, MUC1, MUC5, SOX9, EPCAM, IL6, TGFB1, CXCL10, CD133, UHRF1, and SOX17. This would provide a more comprehensive molecular characterization of the cell lines and allow for better contextualization within the CCA biology landscape.

4. If formalin-fixed, paraffin-embedded tissue samples from the original patient tumors are available (or from another cohort of patients as a validation cohort), performing immunohistochemistry to assess the expression of the corresponding proteins would be highly informative. This would facilitate the correlation of genomic alterations with potential protein-level dysregulation in the tumor tissues.

5. Additionally, should clinical data from the patient donors be accessible, it would be beneficial to include a description of the phenotypic characteristics of each CCA donor sample, such as tumor grade, stage, and clinical outcomes. This would significantly enhance the clinical relevance of the findings.

6. Lastly, the authors could consider performing association analyses between the detected mutations and clinical features of cholangiocarcinoma using publicly available datasets, such as those from The Cancer Genome Atlas (TCGA). Using these data could provide valuable correlations between the mutations identified and key clinical parameters, such as patient survival rates, thereby strengthening the translational impact of the study.

6. PLOS authors have the option to publish the peer review history of their article (what does this mean? ). If published, this will include your full peer review and any attached files.

**Do you want your identity to be public for this peer review?** For information about this choice, including consent withdrawal, please see our Privacy Policy .

Reviewer #1: **Yes: ** Amaya Lopez-Pascual

---

## [Author Response · Author response to Decision Letter 1]

15 Feb 2025

RE: PONE-D-24-41324

Dear Editor-in-Chief,

We are very thankful again to you and the reviewers for the critical reading of our manuscript entitled “Mitochondrial genomic alterations in cholangiocarcinoma cell lines”, by Faipan et al., to PLOS ONE. We appreciate the reviewers’ comments and suggestions, and have incorporated them into the manuscript as follows:

1. To establish the functional significance of the observed mtDNA mutations, the authors should evaluate the expression levels of the genes harboring these mutations through qPCR and confirm protein expression via western blotting. These analyses would provide crucial information on whether the identified mutations result in corresponding changes at the transcriptional or translational levels.

Ans: Thank you for your suggestion. There is the previous study demonstrated that the KKU-213A CCA cell line, which harbors numerous mtDNA SNVs, had ALDH1A3 expression which significantly rises under lactic acidosis (LA) conditions, and correlates with LDHA expression. Higher ALDH1A3 levels have been linked to poorer patient survival, indicating its potential as a prognostic marker, which we have added into the discussion part line 405 to 409 as highlighted in blue (Thamrongwaranggoon et al., 2022). We also assessed the expression of key proteins, including CK19 and IL-6, using Western blotting to investigate potential functional consequences of the observed mtDNA mutations. Our results show that KKU-023, KKU-213A, and KKU-452 exhibit high expression levels of both CK19 and IL-6 proteins, while KKU-055, KKU-100, and MMNK-1 demonstrate lower expression levels (See Response to Reviewer No. 3 below). Moreover, to explore the potential link between mtDNA mutations and protein expression, we analyzed the correlation between CK19 and IL-6 protein expression level and the total number of mtDNA single nucleotide variants (SNVs) and insertions-deletions (INDELs). No significant correlations were found between mtDNA mutations and CK19 protein expression levels, aligning with our literature review, which revealed no prior studies addressing this relationship. In contrast, IL-6 protein expression has been linked to TFB2M, a key regulator of mitochondrial DNA (mtDNA) transcription and maintenance. Overexpression of TFB2M has been shown to increase extracellular mtDNA and IL-6 levels in ovarian cancer cells, promoting M2 macrophage infiltration via the cytoplasmic mtDNA/TLR9/NF-κB/IL-6 pathway (Lei et al., 2024). Although our results did not demonstrate a significant association. We have added this information in the discussion part line 409 to 413 highlighted in blue.

2. The manuscript would benefit from the inclusion of phenotypic characterization of the CCA-derived cell lines. Specifically, conducting colony formation assays and determining the doubling time of each cell line would offer important insights into their proliferative and clonogenic capacities, further contributing to understanding their oncogenic potential.

Ans: Thank you for your insightful suggestion. The doubling time and the total number of mtDNA SNVs and INDELs for each cell line have already been provided in the table below. We also analyzed the correlation between them, unfortunately no significant correlation was observed. We put this sentence in the discussion part as in line 415-417 as highlighted in blue. P value of correlation curve between doubling time and total INDEL and total SNV is 0.6741 and 0.4909, respectively.

3. It is also recommended to assess the expression of cholangiocarcinoma-related genes by qPCR, including markers such as CK19, MUC1, MUC5, SOX9, EPCAM, IL6, TGFB1, CXCL10, CD133, UHRF1, and SOX17. This would provide a more comprehensive molecular characterization of the cell lines and allow for better contextualization within the CCA biology landscape.

Ans: Thank you for your valuable suggestion. We assessed the expression of CK19 and IL-6 at the protein level using western blot analysis. Our results demonstrate that KKU-023, KKU-213A, and KKU-452 cell lines exhibited high CK19 and IL-6 protein expression, while KKU-055, KKU-100, and MMNK-1 showed low expression levels, as illustrated by the protein intensity bar graphs in Figure 3 below. Additionally, we constructed a correlation curve to evaluate the relationship between CK19 and IL-6 protein intensities and the total number of mtDNA single SNVs and INDELs (Figure 3. And 4.). However, this analysis did not reveal significant correlations.We acknowledge the importance of a more comprehensive molecular characterization and will consider expanding our analysis to include the suggested cholangiocarcinoma-related genes in future studies.

4. If formalin-fixed, paraffin-embedded tissue samples from the original patient tumors are available (or from another cohort of patients as a validation cohort), performing immunohistochemistry to assess the expression of the corresponding proteins would be highly informative. This would facilitate the correlation of genomic alterations with potential protein-level dysregulation in the tumor tissues.

Ans: Thank you so much for a great suggestion. We are now conducting the mitochondria mutation analysis in the lager number of cholangiocarcinoma tissues and your suggestion will be absolutely performed.

5. Additionally, should clinical data from the patient donors be accessible, it would be beneficial to include a description of the phenotypic characteristics of each CCA donor sample, such as tumor grade, stage, and clinical outcomes. This would significantly enhance the clinical relevance of the findings.

Ans: Thank you for the suggestion. We have added the table of the clinical characteristics of the patient donors from whom the cholangiocarcinoma (CCA) cell lines as in Table 8 of the revised version.

6. Lastly, the authors could consider performing association analyses between the detected mutations and clinical features of cholangiocarcinoma using publicly available datasets, such as those from The Cancer Genome Atlas (TCGA). Using these data could provide valuable correlations between the mutations identified and key clinical parameters, such as patient survival rates, thereby strengthening the translational impact of the study.

Ans: We respectfully note that TCGA primarily focuses on nDNA data, including somatic and germline mutations, gene expression, and other features related to nuclear genomics. While TCGA has been invaluable for understanding various aspects of cancer biology, it is not well-suited for the detailed investigation of mtDNA mutations. Unfortunately, the TCGA data does not provide direct or comprehensive coverage of mtDNA as its sequencing methods prioritize nuclear genomes. Although some mtDNA reads can be indirectly extracted from whole-genome or whole-exome sequencing data, these reads are often limited, lack sufficient depth, and can result in incomplete or biased mtDNA mutation profiles. Therefore, conducting meaningful association analyses between mtDNA mutations and clinical features using TCGA data would not be feasible or accurate for this study's specific focus on mitochondrial genomics in CCA. Based on this limitation, we have chosen to rely on direct sequencing and analysis of mtDNA from CCA cell lines in our investigation to ensure robust and accurate identification of mtDNA variants. We believe this approach is more appropriate for achieving the study's aims.

We have also added Hasaya Dokduang as the co-author in the revised manuscript because she is the person who handles western blot analysis, which is shown in this response to reviewers’ letter.

We believe that the manuscript has been improved satisfactorily and hope that it is now acceptable for publication in PLOS ONE.

Yours Sincerely,

Watcharin Loilome

---

## [Decision Letter · Decision Letter 1]

26 Feb 2025

PONE-D-24-41324R1Mitochondrial genomic alterations in cholangiocarcinoma cell linesPLOS ONE

Dear Dr. Loilome,

Thank you for submitting your manuscript to PLOS ONE. After careful consideration, we feel that it has merit but does not fully meet PLOS ONE’s publication criteria as it currently stands. Therefore, we invite you to submit a revised version of the manuscript that addresses the points raised by the reviewer.

We look forward to receiving your revised manuscript.

Kind regards,

Matias A Avila, Ph.D.

Academic Editor

PLOS ONE

Journal Requirements:

Reviewers' comments:

Reviewer's Responses to Questions

**Comments to the Author**

1. If the authors have adequately addressed your comments raised in a previous round of review and you feel that this manuscript is now acceptable for publication, you may indicate that here to bypass the “Comments to the Author” section, enter your conflict of interest statement in the “Confidential to Editor” section, and submit your "Accept" recommendation.

Reviewer #1: (No Response)

2. Is the manuscript technically sound, and do the data support the conclusions?

Reviewer #1: Yes

3. Has the statistical analysis been performed appropriately and rigorously? 

Reviewer #1: Yes

4. Have the authors made all data underlying the findings in their manuscript fully available?

Reviewer #1: Yes

5. Is the manuscript presented in an intelligible fashion and written in standard English?

Reviewer #1: Yes

6. Review Comments to the Author

Reviewer #1: Thank you for your detailed responses and thoughtful revisions to the manuscript. I appreciate the effort in incorporating the reviewers' comments and strengthening the discussion. I have a few additional suggestions that would further improve the manuscript:

1. Western Blot Data as Supplementary Material: The Western blot results should be included as supplementary material. Additionally, if not already done, Western blot intensity quantifications should be normalized to β-actin to ensure accurate comparisons. The supplementary material should include the corrected quantification graphs and the corresponding Western blot images. I think it is interesting to have this first molecular characterization. Even if nonsignificant results were found, there are differences in the expression of CK19 specially. the manuscript should mention that further molecular characterization is needed. How many replicates were performed? data are presented as bars with SD or SEM, and is impossible to infer the number of replicates.

2. qPCR Expression Analysis and Correlations: The expression levels of the suggested markers should be evaluated by qPCR. These data should then be used for correlation analyses with doubling time, SNVs, and INDELs. Since qPCR is a straightforward measurement, these additional data would provide valuable support for the characterization, making the findings more robust rather than purely observational.

These additions would significantly strengthen the manuscript and provide a more comprehensive molecular characterization of the studied cell lines. I greatly appreciate the inclusion of Table 8 in response to point 5, as it provides valuable information and significantly enhances the clarity of the results. It is a meaningful addition to the manuscript.

7. PLOS authors have the option to publish the peer review history of their article (what does this mean? ). If published, this will include your full peer review and any attached files.

**Do you want your identity to be public for this peer review?** For information about this choice, including consent withdrawal, please see our Privacy Policy .

Reviewer #1: **Yes: ** Amaya Lopez-Pascual

---

## [Author Response · Author response to Decision Letter 2]

14 Apr 2025

8th April 2025

RE: PONE-D-24-41324R1

Dear Editor-in-Chief,

We are very thankful again to you and the reviewers for the critical reading of our manuscript entitled “Mitochondrial genomic alterations in cholangiocarcinoma cell lines”, by Faipan et al., to PLOS ONE. We appreciate the reviewers’ comments and suggestions, and have incorporated them into the manuscript as follows:

1. Western Blot Data as Supplementary Material: The Western blot results should be included as supplementary material. Additionally, if not already done, Western blot intensity quantifications should be normalized to β-actin to ensure accurate comparisons. The supplementary material should include the corrected quantification graphs and the corresponding Western blot images. I think it is interesting to have this first molecular characterization. Even if nonsignificant results were found, there are differences in the expression of CK19 specially. The manuscript should mention that further molecular characterization is needed. How many replicates were performed? data are presented as bars with SD or SEM and is impossible to infer the number of replicates.

Author’s response: Thank you for your valuable feedback. We have already performed three independent western blot analysis and intensity normalization to β-actin and included the quantification graphs (Figure S8) along with the corresponding representative western blot images (Figures S7) in the supplementary material. The correlation between IL-6 and CK-19 western blotting intensity with doubling time, SNVs, and INDELs are also provided (Figure S9-10) We also provide the raw data of the three independent western blot analysis as below.

2. qPCR Expression Analysis and Correlations: The expression levels of the suggested markers should be evaluated by qPCR. These data should then be used for correlation analyses with doubling time, SNVs, and INDELs. Since qPCR is a straightforward measurement, these additional data would provide valuable support for the characterization, making the findings more robust rather than purely observational.

Author’s response: Thank you for your valuable suggestion. We appreciate your recommendation to include quantitative expression analysis to strengthen our findings.

In response, we have performed droplet digital PCR (ddPCR) to quantitatively assess the expression levels of IL-6, CK19, and β-actin across our cell lines. These data are presented in Table S16. Gene expression levels of IL-6 and CK19 were normalized to β-actin for each cell line (Figure S11). Subsequently, we conducted correlation analyses between the normalized gene expression and key cellular features, including doubling time, SNVs, and INDELs. The corresponding correlation plots are shown in Figures S12 and S13. Although the correlation analyses did not yield statistically significant results, likely due to the limited number of cell lines available, we believe the inclusion of these data adds quantitative depth to our study and provides a basis for future investigations with larger sample sizes.

S16 Table. Droplet digital PCR results of IL-6, CK-19, and β-actin concentration

Sample IL-6 concentration (cp/µL) CK19 concentration (cp/µL) β-actin concentration (cp/µL)

KKU-023 33.35 30.52 6666.7

KKU-055 11.82 3.83 6486.3

KKU-100 125.7 7.722 25500

KKU-213A 3785.5 72.38 24500

KKU-452 77.72 0.426 53100

MMNK-1 4582.6 84.21 73000

S12 Fig. Gene expression level of IL-6 after normalized with β-actin (left above) of each cell line demonstrated by droplet digital PCR. Correlation curves of IL-6 gene expression with doubling time (right above), SNVs (left below), and INDELs (right below), respectively.

S13 Fig. Gene expression level of CK19 after normalized with β-actin (left above) of each cell line demonstrated by droplet digital PCR (left above). Correlation curves of CK19 gene expression with doubling time (right above), SNVs (middle), and INDELs (below), respectively.

We believe that the manuscript has been improved satisfactorily and hope that it is now acceptable for publication in PLOS ONE.

Yours Sincerely,

Watcharin Loilome, Ph.D.

Department of Systems Biosciences and Computational Medicine

Faculty of Medicine,

Khon Kaen University, Khon Kaen, 40002, THAILAND

---

## [Editor Report · Decision Letter 2]

16 Apr 2025

Mitochondrial genomic alterations in cholangiocarcinoma cell lines

PONE-D-24-41324R2

Dear Dr. Loilome,

We’re pleased to inform you that your manuscript has been judged scientifically suitable for publication and will be formally accepted for publication once it meets all outstanding technical requirements.

Kind regards,

Matias A Avila, Ph.D.

Academic Editor

PLOS ONE
---

## [Editor Report · Acceptance letter]

PONE-D-24-41324R2

PLOS ONE

Dear Dr. Loilome,

I'm pleased to inform you that your manuscript has been deemed suitable for publication in PLOS ONE. Congratulations! Your manuscript is now being handed over to our production team.

Kind regards,

on behalf of

Dr Matias A Avila

Academic Editor

PLOS ONE